# Sulfated Triterpene Glycosides from the Far Eastern Sea Cucumber *Cucumaria djakonovi*: Djakonoviosides C_1_, D_1_, E_1_, and F_1_; Cytotoxicity against Human Breast Cancer Cell Lines; Quantitative Structure–Activity Relationships

**DOI:** 10.3390/md21120602

**Published:** 2023-11-22

**Authors:** Alexandra S. Silchenko, Anatoly I. Kalinovsky, Sergey A. Avilov, Roman S. Popov, Ekaterina A. Chingizova, Ekaterina S. Menchinskaya, Elena A. Zelepuga, Elena G. Panina, Vadim G. Stepanov, Vladimir I. Kalinin, Pavel S. Dmitrenok

**Affiliations:** 1G.B. Elyakov Pacific Institute of Bioorganic Chemistry, Far Eastern Branch of the Russian Academy of Sciences, Pr. 100-letya Vladivostoka 159, 690022 Vladivostok, Russia; kaaniv@piboc.dvo.ru (A.I.K.); avilov_sa@piboc.dvo.ru (S.A.A.); popov_rs@piboc.dvo.ru (R.S.P.); chingizova_ea@piboc.dvo.ru (E.A.C.); ekaterinamenchinskaya@gmail.com (E.S.M.); zel@piboc.dvo.ru (E.A.Z.); kalininv@piboc.dvo.ru (V.I.K.); 2Kamchatka Branch of Pacific Institute of Geography, Far Eastern Branch of the Russian Academy of Sciences, Partizanskaya st. 6, 683000 Petropavlovsk-Kamchatsky, Russia; egpanina777@gmail.com (E.G.P.); stepanovvadim24@gmail.com (V.G.S.)

**Keywords:** *Cucumaria djakonovi*, Dendrochirotida, triterpene glycosides, djakonoviosides, sea cucumber, hemolytic and cytotoxic activity, human breast cancer, QSAR

## Abstract

Four new mono- and trisulfated triterpene penta- and tetraosides, djakonoviosides C_1_ (**1**), D_1_ (**2**), E_1_ (**3**), and F_1_ (**4**) were isolated from the Far Eastern sea cucumber *Cucumaria djakonovi* (Cucumariidae, Dendrochirotida), along with six known glycosides found earlier in other *Cucumaria* species. The structures of unreported compounds were established on the basis of extensive analysis of 1D and 2D NMR spectra as well as by HR-ESI-MS data. The set of compounds contains six different types of carbohydrate chains including two new ones. Thus, djakonovioside C_1_ (**1**) is characterized by xylose as the second residue, that was a branchpoint in the pentasaccharide chain. Meanwhile, only quinovose and rarely glucose have been found earlier in pentasaccharide chains branched at C-2 of the second sugar unit. Djakonovioside E_1_ (**3**) is characterized by a tetrasaccharide trisulfated chain, with glucose as the second residue. So, in the series of isolated glycosides, three types of sugars in the second position were presented: the most common, quinovose—in six compounds; glucose—in three substances; and the rare xylose—in one glycoside. The set of aglycones was composed of holostane- and non-holostane-type polycyclic systems; the latter comprised normal and reduced side chains. Noticeably, isokoreoside A (**9**), isolated from *C. djakonovi*, was a single glycoside having a 9(11)-double bond, indicating two oxidosqualenecyclases are operating in the process of the biosynthesis of aglycones. Some of the glycosides from *C. djakonovi*, which were characterized by pentasaccharide branched chains containing one to three sulfate groups, are chemotaxonomic features of the representatives of the genus *Cucumaria*. The assortment of sugar parts of *Cucumaria’s* glycosides was broadened with previously undescribed penta- and tetrasaccharide moieties. The metabolic network of sugar parts and aglycones is constructed based on biogenetic relationships. The cytotoxic action of compounds **1**–**10**, isolated from *C. djakonovi*, against human breast cancer cell lines was investigated along with the hemolytic activity. Erythrocytes were, as usual, more sensitive to the membranolytic action of the glycosides than cancer cells. The triple-negative breast cancer MDA-MB-231 cell line was more vulnerable to the action of glycosides in comparison with the other tested cancer cells, while the MCF-7 cell line was less susceptible to cytotoxic action. Djakonovioside E_1_ (**3**) demonstrated selective action against ER-positive MCF-7 and triple-negative MDA-MB-231 cell lines, while the toxic effect in relation to normal mammary epithelial cells (MCF-10A) was absent. Cucumarioside A_2_-5 (**6**) inhibited the formation and growth of colonies of cancer cells to 44% and tumor cell migration to 85% of the control. Quantitative structure–activity relationships (QSAR) were calculated on the basis of the correlational analysis of the physicochemical properties and structural features of the glycosidic molecules and their membranolytic activity. QSAR revealed the extremely complex nature of such relationships, but these calculations correlated well with the observed SAR.

## 1. Introduction

Sea cucumbers are marine invertebrates belonging to the class Holothuroidea of the phylum Echinodermata. They have wide distributions in all the oceans from shallow waters to significant depths. Sea cucumbers are mainly detritophages, collecting food from superficial sediments through their tentacles. The representatives of the order Dendrochirotida, including *Cucumaria djakonovi*, are filtrators, i.e., they filter solid particles disperged in sea water also using their branched tentacles. For this reason, sea cucumbers have great importance for the ecosystem, providing a sort of cleaning service by recycling and decomposing the substrate. Nowadays, a growing number of marine organisms are being chemically investigated in the search for new biomolecules with pharmacological potential. Sea cucumbers are among them because they produce specific metabolites, triterpene glycosides, which are uncommon in the animal kingdom. Investigations of the triterpene glycosides of diverse representatives of the class Holothuroidea have a long history, beginning from structural studies of the artifact genines obtained as a result of acid hydrolysis, followed by the structure establishment of native compounds [1,2]. The continuation of these researches resulted in the appearance of the modern approaches of the separation of complex mixtures of native metabolites giving individual glycosides, including minor ones, possessing unique structural features [3,4]. All these broaden the fundamental knowledge about the chemical diversity of natural products in general and of triterpenoids and their derivatives in particular. Actually, each new research study of the glycosidic composition of previously uninvestigated species or reinvestigation of species that were previously studied, but using modern techniques and equipment, leads to the finding of novel compounds [4,5]. Fairly recently, the metabolomic studies of the glycosidic profiles of diverse species of holothuroids appeared, providing very useful data on the chemical diversity of glycosides, their distribution, and their content inside the different internal parts of animals, as well as their ecological signal functions [6,7,8,9,10]. The interest of researchers in glycosides has been increased by their diverse and unique biological activity, including anticancer activity against different types of tumors, and immunomodulatory properties [11,12,13]. Hence, some of these substances isolated from *C. djakonovi* demonstrated a promising action against human breast cancer cells, especially against the most aggressive triple-negative MDA-MB-231 cell line, suppressing cells’ viability, inhibiting colony formation and growth, and inhibiting the migration of cells [5]. These data indicate that glycosides could be used in the target therapy of breast cancer. An additional direction of scientific interest in these metabolites, based on the use of them as chemotaxonomic markers, allows for defining the systematic position and resolving taxonomic issues by analysis of the structural features of glycosides. Particularly, on the one hand, the isolation of novel compounds from *C. djakonovi* corroborated the apartness of this species from other closely related *Cucumaria* species; on the other hand, the presence of the same compounds in different species of the genus *Cucumaria* confirmed the specificity of glycosides at the genus level [5,14]. The analysis of the structure–activity relationships of these metabolites shows the significance of the different parts of molecules for the demonstration of bioactivity and their influence on the mechanisms of action of glycosides [15]. The accumulation of such data provides the opportunity to predict glycosides’ bioactivity. So, the calculations of the quantitative structure–activity relationships (QSAR) for the glycosides from *C. djakonovi* are one of the first steps in this prospective direction. As data are accumulated, the combinatorial libraries of glycosides are formed and expanded, and, on the basis of the structural biogenetic rows of aglycones and the carbohydrate chains of glycosides, the pathways of biosynthesis could be traced [4], revealing the sequence of enzymes acting during the formation of glycosides, which is necessary for the regulation of the biosynthesis of these promising compounds in the future. 

As a continuation of the study of the glycosides of *C. djakonovi* collected in Avacha Gulf, the isolation, structure elucidation, and biological activity testing of a series of new triterpene glycosides, djakonoviosides C_1_ (**1**), D_1_ (**2**), E_1_ (**3**), and F_1_ (**4**), and several known compounds were reported. The chemical structures of **1**–**4** were elucidated by the analyses of the ^1^H, ^13^C NMR, 1D TOCSY, and 2D NMR (^1^H,^1^H COSY, HMBC, HSQC, ROESY) spectra as well as the HR-ESI mass spectra. All the original spectra are displayed in Appendix A. The hemolytic activity against human erythrocytes and the cytotoxicity against human breast cancer cell lines MCF-7, T-47D, and triple-negative MDA-MB-231, as well as the non-tumor mammary epithelial cell line MCF-10A, were studied. QSAR analysis was conducted for the series of 20 triterpene glycosides found in *C. djakonovi*.

## 2. Results

### 2.1. Structure Elucidation of Glycosides

The crude glycosidic sum of *Cucumaria djakonovi* (1.379 g) was obtained after the hydrophobic chromatography of the concentrated ethanolic extract on a Polychrom-1 column (powdered Teflon, Biolar, Latvia). Then, it was chromatographed on Si gel columns (CC) with the stepped gradient of the eluents system of CHCl3/EtOH/H2O, in ratios of 100:50:4, 100:75:10, 100:100:17, and 100:125:25, and five fractions were obtained. The fractions 3−5, isolated after the repeated CC with the system of eluents CHCl_3_/EtOH/H_2_O (100:75:10), (100:100:17), and (100:125:25), were subsequently submitted to HPLC on reversed-phase semipreparative columns, Phenomenex Synergi Fusion RP (10 × 250 mm) and Synergi Hydro RP (10 × 250 mm), as well as chiral analytical column Kromasil 3-Cellucoat RP (4.6 × 150 mm), to give 10 individual novel and known glycosides (Figure 1).

The sugar configurations in glycosides **1**–**4** were assigned as *D* on the basis of an analogy with all other known triterpene glycosides from sea cucumbers.

The molecular formula of djakonovioside C_1_ (**1**) was determined to be C_60_H_93_O_30_SNa from the [M_Na_ – Na]^−^ ion peak at *m*/*z* 1325.5451 (calc. 1325.5478) in the (*−*)HR-ESI-MS (Appendix A). The structures of the identical aglycone moieties of djakonoviosides C_1_ (**1**) and E_1_ (**3**) and okhotoside A_2_-1 (**5**) were established by the analysis of their NMR spectra (Table 1, (Appendix A). The holostane-type aglycone (characteristic signals of 18(20)-lactone were observed at δ_C_ 180.2 (C-18) and 85.5 (C-20)) contains 7(8)- (the signals of CH-7 at δ_C_ 120.2 and δ_H_ 5.57 (m) and C-8 at δ_C_ 145.5) and 25(26)-double bonds (the signals of C-25 at δ_C_ 145.4 and CH_2_-26 at δ_C_ 110.8 and δ_H_ 4.72 (m)) and 16β-acetoxy group (the signals at δ_C_ 75.3 (C-16), 170.7 (O**C**OCH_3_), and 21.2 (OCO**C**H_3_). The orientation of this substituent was confirmed by the NOE correlation between H-16 and H-32 as well as by the value of the coupling constant (*J*_16/17_ = 8.7 Hz) [16].

Extensive analysis of the ^1^H, ^13^C NMR, and HSQC spectra of the carbohydrate moiety of djakonovioside C_1_ (**1**) (Table 2; Appendix A) indicated the presence of the pentasaccharide chain with β-glycosidic bonds because five doublets of the anomeric protons at δ_H_ 4.73–5.25 (*J* = 6.8–8.0 Hz) and the signals of the corresponding anomeric carbons at δ_C_ 102.4–105.6 were observed. Analysis of the ^1^H, ^1^H COSY, and 1D TOCSY spectra of each monosaccharide residue started from the anomeric proton, followed by ROESY and HSQC correlations analyses, allowed to determine the monosaccharide composition and the positions of glycosidic bonds. Hence, it was found that the second sugar in the chain is the xylose residue (Xyl2) that is linked to the Xyl1 residue at C-2. The second sugar unit (Xyl2) is bound with another two monosaccharides, being a branchpoint of the chain. The third monosaccharide unit—Glc3—was attached to C-4 Xyl2, and the additional xylose residue (Xyl5) was attached to C-2 Xyl2, causing glycosylation effects (δ_C-4 Xyl2_ 77.9 and δ_C-2 Xyl2_ 82.6). The analysis of the NMR spectra showed that 3-*O*-methylglucose (MeGlc4) was a terminal monosaccharide unit linked to C-3 Glc3. The positions of the glycosidic linkages were corroborated by the ROESY and HMBC correlations between the H-1 Xyl1 and H-3 (C-3) of the aglycone, H-1 Xyl2 and H-2 (C-2) Xyl1, H-1 Glc3 and H-4 (C-4) Xyl2, H-1 MeGlc4 and H-3 (C-3) Glc3, and H-1 Xyl5 and H-2 (C-2) Xyl2 (Table 2). The single sulfate group was attached to the C-4 Xyl1, causing an α-shifting effect of its signal to δ_C_ 76.1, instead of the δ_C_ ~70 observed in non-sulfated compounds.

The (*−*)ESI-MS/MS of **1** (Appendix A) demonstrated the fragmentation of the [M_Na_ − Na]^−^ ion, with *m*/*z* 1325.5 giving fragment ion peaks at *m*/*z* 1265.5 [M_Na_ − Na − CH_3_COOH]^−^, 1223.5 [M_Na_ − Na − NaSO_3_ + H]^−^, 1193.5 [M_Na_ − Na − Xyl]^−^, 987.4 [M_Na_ − Na − MeGlc – Glc + H]^−^, 813.2 [M_Na_ − Na – Agl − H]^−^, 681.1 [M_Na_ − Na – Agl − Xyl]^−^, and 595.2 [M_Na_ − Na − Agl − XylSO_3_]^−^. The (*+*)ESI-MS/MS of **1** demonstrated the fragmentation of the [M_Na_ + Na]^+^ ion, with *m*/*z* 1371.5 leading to ion peaks with *m*/*z* 1251.6 [M_Na_ + Na − NaHSO_4_]^+^ and 1179.6 [M_Na_ + Na − MeGlc + H]^+^.

These data indicate that djakonovioside C_1_ (**1**) is 3β-*O*-{3-*O*-methyl-β-D-glucopyranosyl-(1→3)-β-D-glucopyranosyl-(1→4)-[(1→2)-β-D-xylopyranosyl]-β-D-xylopyranosyl-(1→2)-4-*O*-sodium sulfate-β-D-xylopyranosyl}-16β-acetoxyholosta-7,25-dien.

The molecular formula of djakonovioside D_1_ (**2**) was determined as C_61_H_95_O_31_SNa from the [M_Na_ − Na]^−^ ion peak at *m*/*z* 1355.5596 (calc. 1355.5584) and the [M_Na_ − Na − H]^2−^ ion peak at *m*/*z* 677.2767 (calc. 677.2755) in the (*−*)HR-ESI-MS (Appendix A). The aglycone of djakonovioside D_1_ (**2**) (Table 3, Appendix A) was structurally close to that of **1**, only differing by the position of the double bond in the side chain. So, all the signals of the polycyclic nuclei in the NMR spectra of **2** and **1** were almost coincident. The signals of the side chain were assigned by the analysis of the NMR spectra of **2**: an isolated spin system formed by the protons from H-22 to H-24 was found in the ^1^H,^1^H COSY spectrum, and the characteristic signals in the ^13^C and ^1^H NMR spectra at δ_C_ 123.9 (C-24), δ_H_ 5.00 (m, H-24) and 132.1 (C-25) corresponded to the 24(25)-double bond. Its position was confirmed by the correlations H-26(27)/C: 24, 25, 27(26) in the HMBC spectrum.

Extensive analysis of the ^1^H, ^13^C NMR, ^1^H,^1^H COSY, 1D TOCSY, HSQC, and ROESY spectra of the carbohydrate part of djakonovioside D_1_ (**2**) and okhotoside A_2_-1 (**5**) ( Table 4 and Appendix A) revealed the presence of the same monosulphated pentasaccharide chains. The typical signals for quinovose residue were absent, while three signals characteristic for the hydroxymethylene groups of glucopyranose residues at δ_C_ 61.4, 61.1, and 61.7 were observed. Further analysis of the sugar composition and sequence, as well as the glycosidic bond positions, showed that the second and third units in the chain were glucose residues (Glc2 and Glc3); the 3-*O*-methylglucose (MeGlc4) and xylose (Xyl5) residues were terminal, forming a carbohydrate chain branched by C-2 Glc2. A sulphate group was attached to C-4 Xyl1 (δ_C_ 76.1), as in djakonovioside C_1_ (**1**).

The fragment ion peaks in the (*−*)ESI-MS/MS of **2** (Appendix A) were observed at *m*/*z* 1296.5 [M_Na_ − Na − CH_3_COOH]^−^, 1105.5 [M_Na_ − Na − SO_3_Na − Xyl + H]^−^, 843.2 [M_Na_ − Na – Agl − H]^−^, and 723.3 [M_Na_ − Na – Agl − NaSO_4_]^−^ due to fragmentation of the [M_Na_ − Na]^−^ ion with *m*/*z* 1355.6. The ion peak at *m*/*z* 589.2 [M_Na_ − Na − MeGlc]^2−^ appeared as a result of the fragmentation of the [M_Na_ − Na − H]^2−^ ion at *m*/*z* 677.3. The (*+*)ESI-MS/MS of **2** demonstrated the fragmentation of the [M_Na_ + Na]^+^ ion with *m*/*z* 1401.5, giving ion peaks at *m*/*z* 1281.6 [M_Na_ + Na − NaHSO_4_]^+^ and 1209.6 [M_Na_ + Na − MeGlc + H]^+^.

Thus, djakonovioside D_1_ (**2**) is 3β-*O*-{3-*O*-methyl-β-D-glucopyranosyl-(1→3)-β-D-glucopyranosyl-(1→4)-[(1→2)-β-D-xylopyranosyl]-β-D-glucopyranosyl-(1→2)-4-*O*-sodium sulfate-β-D-xylopyranosyl}-16β-acetoxyholosta-7,24-dien.

The molecular formula of djakonovioside E_1_ (**3**) was determined as C_56_H_85_O_33_S_3_Na_3_ from the [M_3Na_ − Na]^−^ ion peak at *m*/*z* 1427.3942 (calc. 1427.3936), the [M_3Na_ − 2Na]^2−^ ion peak at *m*/*z* 702.2037 (calc. 702.2022), and the three-charged ion [M_3Na_ − 3Na]^3−^ at *m*/*z* 460.4732 (calc. 460.4717) in the (*−*)HR-ESI-MS (Appendix A) that confirmed the presence of three sulfate groups. In the ^1^H and the ^13^C NMR spectra of the carbohydrate chain of djakonovioside E_1_ (**3**) (Table 5, Appendix A), the four doublets of the anomeric protons at δ_H_ 4.68–5.09 (*J* = 7.3–8.5 Hz) and the signals of the corresponding anomeric carbons at δ_C_ 103.8–104.7 were indicative of a tetrasaccharide chain with β-glycosidic bonds between sugars. The first monosaccharide connected to the C-3 of the aglycone was a xylose (Xyl1) sulfated by C-4 (deduced from the deshielding of this signal to δ_C_ 76.1). The subsequent analysis of the 1D TOCSY, ^1^H,^1^H COSY, HSQC, and ROESY spectra revealed that the second residue was a glucose, which was linked to C-2 Xyl1. The third sugar—glucose (Glc3)—was attached to the typical position—C-4 Glc2 (cross-peak H-1 Glc3/H-4 Glc2 in the ROESY spectrum)—and was sulphated by C-6 (deshielding of the signal of C-6 Glc3 to δ_C_ 67.4). Terminal 3-*O*-methylglucose residue was bound to C-3 Glc3, that was deduced from the corresponding NOE correlation (Table 5) and also contained a sulphate group because the signal of C-6 MeGlc4 was observed at δ_C_ 67.0. Hence, the tetrasaccharide chain of **3** was new, having glucose as the second unit and bearing three sulfate groups.

The fragment ion peaks in the (*−*)ESI-MS/MS of **3** (Appendix A) were observed as a result of the fragmentation of the [M_3Na_ − Na]^−^ ion at *m*/*z* 1427.5, giving ions at *m*/*z* 1367.4 [M_3Na_ − Na − CH_3_COOH]^−^, 1307.4 [M_3Na_ − Na − NaHSO_4_]^−^, 1029.4 [M_3Na_ − Na − NaHSO_4_–MeGlcSO_3_ + H]^−^, 915.1 [M_3Na_ − Na − Agl − H]^−^, 681.1 [M_3Na_ − Na − Agl − XylSO_3_ − H]^−^, and 519.0 [M_3Na_ − Na − Agl − XylSO_3_ − Glc − H]^−^. The ion peak at *m*/*z* 446.0 [M_3Na −_ 2Na − Agl − H]^2−^ appeared as a result of the fragmentation of the [M_3Na_ − 2Na]^2−^ ion at *m*/*z* 704.2. 

Thus, djakonovioside E_1_ (**3**) is 3β-*O*-[6-*O*-sodium sulfate-3-*O*-methyl-β-D-glucopyranosyl-(1→3)-6-*O*-sodium sulfate*-*β-D-glucopyranosyl-(1→4)-β-D-glucopyranosyl-(1→2)-4-*O*-sodium sulfate-β-D-xylopyranosyl]-16β-acetoxyholosta-7,25-dien.

The molecular formula of djakonovioside F_1_ (**4**) was determined as C_59_H_93_O_34_S_3_Na_3_ from the [M_3Na_ − Na]^−^ ion peak at *m*/*z* 1487.4467 (calc. 1487.4511), the [M_3Na_ − 2Na]^2−^ ion peak at *m*/*z* 732.2320 (calc. 732.2310), and the [M_3Na_ − 3Na]^3−^ ion peak at *m*/*z* 480.4926 (calc. 480.4909) in the (−)HR-ESI-MS (Appendix A). Noticeably, for the isotope composition of the pseudomolecular ion of **4**, the predominance of the [M_3Na_ − Na + 2]^−^ ion peak is inherent. A normal isotope distribution was observed for two- and three-charged pseudomolecular ions. This is obviously explained by the chemical structure of the side chain. The protons of the methylene group CH_2_-23 adjacent to the 22-oxo group and the 24(25)-double bond were easily exchanged to deuterium when the glycoside was dissolved in the mixture C_5_D_5_N/D_2_O for the NMR spectra acquisition. The signal of CH_2_-23 could not be accumulated in the ^13^C NMR spectrum of **4** for the same reason. Therefore, the spectra were repeatedly acquired in C_5_D_5_N/H_2_O, which resulted in the appearance of the signal at δ_C_ 37.0 that corresponded to the proton’s signal at δ_H_ 3.61 (m, H-23) in the HSQC spectrum.

The aglycone of djakonovioside F_1_ (**4**) (Table 6; Appendix A) did not contain a γ-lactone ring (deduced from the absence of the characteristic signal in the ^13^C NMR spectrum at δ_C_ ~180 (C-18)) being of the non-holostane type. Actually, the signals of CH_3_-18 were observed at δ_C_ 24.5 and δ_H_ 1.29 (s). The signals in the downfield region of the ^13^C and ^1^H NMR spectra at δ_C_ 122.1 (C-7), δ_H_ 5.61 (m, H-7), and 148.5 (C-8) were characteristic of the intranuclear double bond, while the signals at δ_C_ 117.3 (C-24), δ_H_ 5.46 (m, H-24), and δ_C_ 134.9 (C-25) indicated the availability of a normal side chain with the 24(25)-double bond in **4**. The signals of the side chain were deduced starting from the signal for C-22. The signal of C-22 was assigned on the basis of HMBC correlations of H-21/C: 20, 22 those are typical for lanostane derivatives. Hence, the signal of quaternary carbon C-22 was observed at δ_C_ 216.1, indicating the presence of an oxo-group. The position of the double bond was assigned as 24(25) based on the HMBC correlations of H-26(27)/C: 24, 25. The (20*R*)-configuration, the same as in lanostane derivatives from sea cucumbers having an oxygen-containing substituent at C-22 [17], was determined on the basis of NOE-correlations H-21/H-12 and H-21/H-17.

The ^1^H and ^13^C NMR spectra of the carbohydrate part of djakonovioside F_1_ (**4**) (Table 7; Appendix A) were characteristic for the pentasaccharide chain (five doublets of anomeric protons at δ_H_ 4.71–5.20 and corresponding to the signals of the anomeric carbons at δ_C_ 102.1–105.2) with β-glycosidic bonds (coupling constants of anomeric protons *J* = 7.2–8.1 Hz). The monosaccharide composition deduced from the analysis of the 1D TOCSY, ^1^H,^1^H COSY, HSQC, and ROESY spectra was two xylose (Xyl1 and Xyl5), quinovose (Qui2), glucose (Glc3), and 3-*O*-methylglucose (MeGlc4) residues. Analysis of the ROESY and HMBC spectra of **4** revealed that the quinovose was a branchpoint of the chain because Xyl5 attached to C-2 Qui2. The rest of the glycosidic bonds occupied typical positions for the glycosides of the sea cucumbers, demonstrating the glycosylation effects in the NMR spectra: δ_C-2 Xyl1_ 81.5, δ_C-4 Qui2_ 86.4, and δ_C-3 Glc3_ 86.6. Three sulphate groups were present in the sugar chain of **4** in the following positions, which were deduced on the basis of α-shifting effects: C-4 Xyl1 (δ_C_ 76.1), C-6Glc3 (δ_C_ 67.3), and C-6 MeGlc4 (δ_C_ 66.4). The same carbohydrate chain composed the molecules of isokoreoside A (**9**) (Appendix A) and koreoside A (**10**) (Appendix A).

The fragment ion peaks in the (*−*)ESI-MS/MS of **4** (Appendix A) were observed as a result of the fragmentation of the [M_3Na_ − Na + 2]^−^ isotopic ion at *m*/*z* 1489.5, giving the ions at *m*/*z* 1211.4 [M_3Na_ − Na + 2−MeGlcSO_3_]^−^, 947.4 [M_3Na_ − Na + 2 − MeGlcSO_3_ − GlcSO_3_]^−^, and 797.1 [M_3Na_ − Na + 2 − MeGlcSO_3_ − GlcSO_3_ − XylSO_3_ − H]^−^, arising as a result of the sequential loss of monosaccharide units. 

Thus, djakonovioside F_1_ (**4**) is 3β-*O*-{6-*O*-sodium sulfate-3-*O*-methyl-β-D-glucopyranosyl-(1→3)-6-*O*-sodium sulfate*-*β-D-glucopyranosyl-(1→4)-[(1→2)-β-D-xylopyranosyl]-β-D-quinovopyranosyl-(1→2)-4-*O*-sodium sulfate-β-D-xylopyranosyl}-(20*R*)-hydroxy-22-oxo-lanosta-7,24-dien.

The structures of known compounds, okhotoside A_2_-1 (**5**) from *C. okhotensis* [18], cucumarioside A_2_-5 (**6**) from *C. conicospermium* [19], frondoside A_2_-3 (**7**) from *C. frondosa* [20], and koreoside A (**10**) from C. *koreaensis* [21], were identified by extensive analysis of the 1D and 2D NMR spectra and compared with the literature data. All the original spectra and the assignments of the signals are provided in Appendix A.

Isokoreoside A (**9**) (Appendix A) was first isolated as a desulfated derivative from the fraction of the trisulfated glycosides of *C. conicospermium* [19], which was separated into individual compounds after the procedure of solvolytic desulfation. In native form, this compound was later obtained from *C. frondosa* [17].

Cucumarioside A_3_-2 (**8**) was isolated as native glycoside from *C. djakonovi* for the first time. The structure of **8** was determined earlier on the basis of its desulfated derivative, obtained the same way as **9** from *Cucumaria conicospermium* [19]. So, the 2D NMR spectra and the assignments of the signals of **8** are provided first (Table 8 and Table 9, Appendix A).

### 2.2. Biologic Activity of the Glycosides

The cytotoxic activity of the compounds isolated from *C. djakonovi* was studied against three types of human breast cancer cells (MCF-7, T-47D, and triple negative MDA-MB-231), as well as the non-tumor mammary epithelial cell line MCF-10A. Djakonovioside A_1_ [5] and cisplatin were used as the positive controls. Cytotoxic activity against all the selected cell lines was assessed using the MTT method (Table 10).

Djakonovioside F_1_ (**4**), okhotoside A_2_-1 (**5**), and cucumarioside A_2_-5 (**6**) showed strong hemolytic activity against human erythrocytes, with ED_50_ 0.51 ± 0.01, 1.53 ± 0.14, and 1.63 ± 0.13 µM, respectively. The hemolytic activity of djakonoviosides C_1_ (**1**) and D_1_ (**2**) was slightly lower but significant and close to each other. Djakonovioside D_1_ (**2**), cucumarioside A_3_-2 (**8**), and isokoreoside A (**9**) demonstrated moderate hemolytic activity but were not active against all human tumor cell lines. Frondoside A_2_-3 (**7**) and koreoside A (**10**) did not show hemolytic or cytotoxic activity in the concentration range up to 50 µM.

The estimation of the selectivity index (Table 11) showed djakonovioside E_1_ (**3**) was a leader, demonstrating the strongest cytotoxicity against the MCF-7 cell line (IC_50_ 1.52 ± 0.14 μM) as well as against the triple negative MDA-MB-231 cell line (IC_50_ 2.19 ± 0.17 µM). At the same time, this glycoside was not toxic in relation to normal mammary epithelial cells (MCF-10A). None of the other glycosides showed similar selectivity. But the MDA-MB-231 cell line was more sensitive to their action compared with the T-47D and, especially, MCF-7 cell lines.

The cytotoxic activity of djakonovioside E_1_ (**3**) was maximal in the series against the MCF-7 and MDA-MB-231 cell lines with a half-maximal inhibitory concentration of 1.52 ± 0.14 μM (Figure 2a) and 2.19 ± 0.17 µM (Table 10), respectively. Cucumarioside A_2_-5 (**6**) was the most active compound from the series in relation to T-47D cells (IC_50_ 5.81 ± 0.86 μM (Figure 2b). Djakonovioside C_1_ (**1**) and cucumarioside A_2_-5 (**6**) demonstrated a pronounced effects against the MDA-MB-231 cell line (IC_50_ of 7.67 ± 0.32 and 2.58 ± 0.1 μM, respectively) (Figure 2c,d).

To study antiproliferative properties, three most active glycosides were selected: djakonoviosides C_1_ (**1**) and E_1_ (**3**) and cucumarioside A_2_-5 (**6**). The prolonged incubation of cells for 48 and 72 h with glycosides **1** and **6** did not increase their EC_50_; but, more importantly, the glycosides did not lose cytotoxicity over time. Only djakonovioside E_1_ (**3**) showed an antiproliferative effect when incubated with MCF-7 cells for 48 h, demonstrating approximately a two-fold increase in EC_50_ (0.78 ± 0.32 μM) (Figure 2a).

The clonogenic (or colony formation) assay is a standard in vitro cell survival assay based on the ability of a single cell to grow into a colony. In the human body, this uncontrolled growth of tumor cells leads to the formation of metastases. To study the effect of selected glycosides on the formation and growth of tumor cell colonies, a range of non-toxic concentrations was used against the MDA-MB-231 cell line (Figure 3a). Additionally, colonies of MCF-7 lines were exposed to the action of non-toxic doses of djakonovioside E_1_ (**3**) (Figure 3b). For all tested compounds, a dose-dependent effect of inhibiting colony growth was observed. Cucumarioside A_2_-5 (**6**) at a concentration of 1 μM demonstrated the greatest inhibitory effect on the formation and growth of colonies: 44.32 ± 0.77% of the control. The inhibitory effects of djakonovioside C_1_ (**1**) at concentrations of 2 and 1 µM were the same—approximately 25% compared to the control. Djakonovioside E_1_ (**3**) blocked the formation of colonies of both MDA-MB-231 and MCF-7 to the same extent: at a concentration of 2 μM in relation to MDA-MB-231 cells to 35.68 ± 2.00% of the control and in relation to MCF-7 cells to 30.73 ± 0.49% of the control (Figure 3a,b).

Scratch analysis is used to study the effects of compounds with potential antitumor activity on cell motility and cell–cell interactions. In the control group of the MDA-MB-231 cell line, the scratch was overgrown within 24 h, while in the control group of the MCF-7 cell line, this happened after 72 h (Figure 4e). All the selected glycosides in the concentration range below their EC_50_ inhibited the migration of breast cancer MDA-MB-231 and MCF-7 cells in a dose-dependent manner. The greatest effect, about 85% as compared to the control, was observed for cucumarioside A_2_-5 (**6**) at a concentration of 1 μM after 24 h of incubation with MDA-MB-231 cells (Figure 4b). Images of CFDA SE fluorescently labeled MCF-7 cells incubated with djakonovioside E_1_ (**3**) demonstrate the reliable blockage of the migration of cells of the MCF-7 line under the glycoside action (Figure 4e).

### 2.3. Correlational Analysis and QSAR Model

The quantitative structure–activity relationship (QSAR) approach was applied to analyze the correlation between the hemolytic activity values and structures of all the glycosides isolated from *C. djakonovi*. Three-dimensional models of the glycosides were built, protonated at pH 7.4, and subjected to energy minimization. A conformational search was performed with MOE 2020.0901 CCG software [22], and the dominant glycoside conformations were selected for further analysis. A set of various 2D and 3D descriptors (379 in total) responsible for physicochemical properties, as well as energy values and topological indexes numerically expressing the geometric properties of molecular structures, was calculated and analyzed using the QuaSAR-Descriptor tool of the MOE 2020.0901 CCG software [22].

Noticeably, the descriptors choice has a fundamental significance, since no “almighty descriptor set” modeling all the activities and properties has been found yet. So, the selection of a suitable descriptor set for each activity and type of analyzed compounds is needed. So, in addition to the descriptors characterizing the physicochemical properties of the molecules (polarizability, refractive index, surface charge distribution, dipole moment, hydrogen bonds’ potential strength (donors and acceptors) [23], hydrophobic volume, surface area, atomic valence connectivity index, etc.), following descriptors as the presence/absence of 18(20)-lactone and the side chain, carbohydrate chain branching, and nature of the second sugar residue (glucose, quinovose, and xylose), the sulfate groups’ number and positions were added to the descriptors set provided by the MOE-QuaSAR-Descriptor software (2020.0901 CCG). The correlational analysis revealed the direct positive correlation between the hemolytic activities of the tested compounds in vitro and such descriptors as the molecular refractivity, log of the octanol/water partition coefficient [24], and partial charges distribution on the van der Waals surface area. In contrast, the diameter of the molecule, principal moment of inertia describing the different aspects of molecular shape, VDW surface area (Å^2^), molecular VDW volume (Å^3^), lowest hydrophobic energy, and approximation of the sum of the VDW surface areas of hydrophobic atoms (Å^2^) were found to negatively correlate.

The analysis of the principal components (PCA) decreased the number of descriptors, leaving only those having a substantial contribution, and resulted in the division of the glycosides into two groups (Figure 5), which indicated the right way of the descriptor’s choice. The linear QSAR model was built with the QuaSAR-Model tool of the MOE 2020.0901 CCG software [22] using these descriptors. The model fits well with the experimental data on the hemolytic activities of glycosides with a correlation coefficient r^2^ = 0.94702 and RMSE = 0.05234 (Appendix A). The model was cross-validated with r^2^_cros_ = 0.82341 and RMSE_cros_ = 0.24613. The QSAR model includes 148 terms, 58 from those that have the biggest contribution; however, the reduction in the number of descriptors to the latter value resulted in the quality deterioration of the correlation model. All these data indicate the extremely complex nature of the relationships between the structure of glycosides and their membranolytic action, with the multiple negligible effects of plenty of descriptors causing a considerable effect in combination with each other. 

## 3. Discussion

### 3.1. Analysis of Structural Peculiarities of Glycosides: Significance for Chemotaxonomy and Biogenesis

Six compounds (**1**–**3** and **5**–**7**) from the last series of glycosides isolated from *C. djakonovi* contained four different holostane-type aglycones. The aglycone of new djakonoviosides C_1_ (**1**), and E_1_ (**3**) and known okhotoside A_2_-1 (**5**) was found earlier in the glycosides of sea cucumbers *C. japonica* [25], *Neothyonidium magnum* [26], and *Thyone aurea* [27]. The aglycone of djakonovioside D_1_ (**2**) is present in plenty of glycosides from different species of sea cucumbers [15,28,29,30,31,32,33]. The aglycone of known cucumarioside A_2_-5 (**6**) was also revealed in okhotoside A_1_-1 [18] and cucumarioside A_0_-1 [34] isolated earlier from *C. djakonovi* [5]. The aglycone of frondoside A_2_-3 (**7**)—the glycoside of *C. frondosa* [20]—characterized by the absence of any substituents at C-16 as well as by the differing side chain structure is not so common. It was repeatedly found only in chitonoidoside K_1_ from *Psolus chitonoides* [35]. All these aglycones are characterized by the presence of a 7(8)-double bond, being the products of one oxidosqualenecyclase (OSC).

Four reported glycosides (**4** and **8**–**10**) contained non-holostane aglycones without a lactone ring, including a new one in djakonovioside F_1_ (**4**). Noticeably, djakonovioside F_1_ (**4**) is the only compound from this row having a normal non-shortened side chain. The other three aglycones are hexa-*nor*-lanostane derivatives. Djakonovioside F_1_ (**4**), cucumarioside A_3_-2 (**8**), and koreoside A (**10**) formed a biogenetic row reflecting the formation in the process of the biosynthesis of nor-lanostane derivatives through the stage of C-22 oxidation and following the oxidative cleavage of the 20(22)-covalent bond. This leads to the elimination of a side chain. The same way is realized during the biosynthesis of steroid hormones. Remarkably, the set of trisulfated pentaosides of *C. djakonovi* (djakonovioside F_1_ (**4**), isokoreoside A (**9**), and koreoside A (**10**)) complements and resembles the analogical set found in *C. frondosa* [17]: three pairs of isomers differed by the positions of the double bonds in the lanostane nuclei had a differently substituted C-22 or a shortened side chain. Compound **4**, having a carbonyl group at C-22, can be considered as a missing link in the row of aglycones of the *C. frondosa* glycosides that fills the gap between the 22-hydroxylated (22-*O*-acetylated) and hexa-*nor*-lanostane derivatives.

Interestingly, isokoreoside A (**9**) is the single glycoside from *C. djakonovi* having a 9(11)-double bond. This indicates that two oxidosqualenecyclases (OSCs) (parkeol syntase and 9βH-lanosta-7,24-diene-3β-ol syntase) are operating in the biosynthesis and forming different types of polycyclic nuclei of the glycosides of *C. djakonovi*. However, the parkeol syntase seems to be preferably engaged in the biosynthesis of free 14α-methyl sterols with a 9(11)-double bond, while the 9βH-lanosta-7,24-diene-3β-ol synthase’s action leads to triterpene aglycones’ formation. Recent investigations into sea cucumber glycosides showed that some species contain mainly one type of aglycones with a certain position of intranuclear double bond, but the others accumulate glycosides with different positions for the double bonds in the polyclic system [4]. However, even when glycosides preferably contain aglycones with one position of the intranuclear double bond, for example, Δ^7(8)^-aglycones in *E. fraudatrix* [36] and *S. horrens* [37] and Δ^9(11)^-aglycones in *A. japonicus* [38], the genes of at least two OSCs are expressed, albeit with different efficiencies. In *C. djakonovi*, the situation seems to be different: the level of expression of parkeol syntase and its activity is obviously on the high level; however, the major part of biosynthesized parkeol is subsequently used for the formation of 14α-methylsterols with a 9(11)-double bond. Such parkeol-derived sterols are very typical for sea cucumbers belonging to the order Dendrochirotida, including the family Cucumariidae [39,40,41,42,43].

Generally, five new aglycones were found in the glycosides isolated from *C. djakonovi* [5]. Noticeably, the new aglycones are mainly inherent for monosulfated compounds, while, among trisulfated compounds, only one glycoside having a novel aglycone was discovered. Eight types of carbohydrate chains comprised the glycosides of *C. djakonovi*. Two sugar chains, know earlier from the glycosides of *C. okhotensis* [18], *C. japonica* [34], and *C. frondosa* [44,45], are the parts of the djakonoviosides of groups A and B (monosulphated tetra- and pentaosides, with second quinovose and third xylose residues). Among the last series of glycosides from *C. djakonovi*, two types of carbohydrate moieties were found for the first time, while four were known earlier. The monosulfated pentasaccharide chain of djakonovioside C_1_ (**1**), with xylose as the second unit, was new. Such a structural feature is very rare for holothurious glycosides. It was only found in two of several hundred compounds known from representatives of the order Dendrochirotida [46,47]. Additionally, the trisulfated tetrasaccharide chain of djkonovioside E_1_ (**3**), with glucose as the second unit, was revealed first. The oligosaccharide parts of djakonoviosides D_1_ (**2**) and F_1_ (**4**) and known compounds **5**–**10** are also characteristic for other representatives of the *Cucumaria* genus: *C. okhotensis* [18], *C. conicospermium* [19], *C. frondosa* [17,20], *C. japonica* [48], and *C. koreaensis* [21]. These data confirm the possibility of using glycosides as chemotaxonomic markers. Belonging to *C. djakonovi* to the genus *Cucumaria* is undoubtfully evidenced by the presence of the same glycosides, characteristic to the other species of the genus. The supposition that all *Cucumaria* species share the same mono-, di-, and trisulfated pentasaccharide branched chains and species-specific aglycones is corroborated by the structures of *C. djakonovi* glycosides [14]. However, the set of sugar chains biosynthesized by sea cucumbers of the *Cucumaria* genus is broadened with novel oligosaccharide moieties.

Biogenetic analysis of sugar chain structures of *C. djakonovi* glycosides showed that each type of carbohydrate chain is formed by a single way, including glycosylation with a certain monosaccharide residue or sulfation reactions. The branching of pathways occurs at the stage of glycosylation of the monoxylosides with additional xylose (forming the djakonovioside C_1_ (**1**) chain), quinovose, or glucose residues (Figure 6). Subsequently the pathway of the biosynthesis of quinovose-containing glycosides divides depending on the type of third monosaccharide (xylose or glucose), which glycosylates quinovose residue by C-4. At this stage, tetrasaccharide chains of the djakonoviosides of group A are formed. Next, branching with xylose leads to the djakonoviosides of group B. In the case of glycosides having glucose as the third sugar, the tetrasaccharide precursors are obviously initially subjected to glycosylation, forming the pentasaccharide chains of cucumarioside A_2_-5 (**6**) and frondoside A_2_-3 (**7**), followed by further sulfation leading to disulfated cucumarioside A_3_-2 (**8**) and trisulfated djakonovioside F_1_ (**4**), isokoreoside A (**9**), and koreoside A (**10**).

The third direction of sugar chains’ biosynthesis is realized when the glucose glycosylates monoxylosides. The next steps of the chain’s elongation are glycosylation with glucose and 3-*O*-methylglucose (third and fourth residues). Then, two pathways are possible: the two stages of sulfation resulting in the chain of djakonovioside F_1_ (**4**) or the glycosylation of the C-2 position of Glc2 leading to the chain of djakonovioside D_1_ (**2**).

So, the biosynthesis of the carbohydrate moieties of *C. djakonovi* glycosides looks rather strictly directed, in comparison with that of the aglycones, exhibiting the time shifts of some stages that is typical for the mosaic type of biosynthesis. The mosaicism of the biosynthesis of the glycosides of *C. djakonovi* clearly appeared at the level of the whole molecules as the combination of different sugar moieties with the same aglycones, due to the parallel and independent biosynthesis of these parts of molecules. There are some examples: cucumariosides A_2_-5 (**6**) and A_0_-1 and okhotoside A_1_-1 [5], as well as djakonoviosides C_1_ (**1**) and E_1_ (**3**) and okhotoside A_2_-1 (**5**), share the same aglycone.

It is interesting that hexa-*nor*-lanostane glycosides, structurally similar to sex (steroid) hormones, are mainly trisulfated or rarely disulfated (the biosynthetic precursors of trisulfated) compounds in the representatives of the genus *Cucumaria* [17,19,21], including *C. djakonovi*. These data, along with the metabolite profiling of a representative of the family Sclerodactylidae—the sea cucumber *Eupentacta fraudatrix* [10], where *nor*-lanostane derivatives were quantitatively predominant in the gonads—probably indicate that the role of such metabolites is different from that of the other glycosides, which provide the chemical defense of sea cucumbers. Actually, it is known that glycosides also synchronize the oocyte maturation in the holothurious population. So, a high level of sulfation makes them more polar and hydrophilic and facilitates their release into the surrounding sea water. Such an exchange with chemical signals provides the simultaneous readiness to spawn all the animals in the population.

### 3.2. Tendencies of Biologic Activity of the Glycosides: Observed and Calculated Structure Activity Relationships (SAR and QSAR)

The observed structure–activity relationships based on the hemolytic activity demonstrated that the most active compounds, okhotoside A_2_-1 (**5**) and cucumarioside A_2_-5 (**6**), from the tested series have holostane-type aglycones and monosulphated pentasaccharide chains. The high hemolytic effect of djakonovioside F_1_ (**4**) was unexpected due to the absence of a lactone in **4** and the presence of three sulfate groups. The presence of a 22-keto group in the side chain of **4** in close proximity to the 20-OH group obviously resulted in the intramolecular hydrogen bond formation leading to a spatial structure similar to the lactone ring that resulted in the increase of the activity. Djakonovioside C_1_ (**1**), having the same aglycone as okhotoside A_2_-1 (**5**), showed slightly lower activity due to the presence of xylose as the second residue of the sugar chain. Djakonovioside D_1_ (**2**), being isomeric to **5** by the double bond position in the side chain, also was slightly less hemolytic. The absence of the activity of frondoside A_2_-3 (7) is easily explained by the presence of a hydroxy group in its side chain [15]. The low hemolytic activity of cucumarioside A_3_-2 (**8**) and isokoreoside A (**9**) is determined by the presence of non-holostane aglycones with shortened side chains; the same features led to the complete loss of the activity of koreoside A (**10**). The partial compensation of the hemolytic action of **8** and **9**, in comparison with compound **10**, was presumably realized due to the presence of two sulfate groups instead of three in **8** and the 9(11)-position of the intranuclear double bond in **9**. Erythrocytes were, as usual, more sensitive to the membranolytic action of glycosides than cancer cells.

Quantitative structure–activity relationships were calculated on the basis of correlational analysis of the physicochemical properties and structural features of the glycosidic molecules and their membranolytic activity and revealed the extremely complex nature of such relationships. The characteristics of molecules related with charged groups, such as polarization under the influence of the electric fields of neighboring ions, surface charge distribution, and impact of hydrophobic/hydrophilic areas and some others considerably influence the membranotropic activity of glycosides. The discovered dependence of the activity upon such chemical peculiarities as the number and positions of double bonds, single bond chain (aglycone side chain) length, availability of 18(20)-lactone, branching and monosaccharide composition of the carbohydrate chain, and positions and numbers of sulfate groups logically follows from the above physical characteristics. More importantly, the QSAR results are consistent with the observed structure–activity relationships. Actually, the availability of a normal non-shortened side chain is urgently needed for the glycoside to be active (i.e., hexa-nor-lanostane compounds **8**–**10** are almost not active). The presence of 18(20)-lactone also provides significant activity, as illustrated not only by *C. djakonovi* glycosides [5] but also is a common tendency [15]. The negative correlation of the molecular volume and shape with the hemolytic activity is confirmed by the observation that tetraosides with linear carbohydrate chains showed stronger effects than the corresponding pentaosides [5,15]. The calculations showed that the number of sulfate groups has an ambiguous effect on the activity of the tested glycosides. In this case, the calculations are also backed with observations: disulfated cucumarioside A_3_-2 (**8**) is more active than trisulfated koreoside A (**10**). The presence of a third sulfate group, unlike the second one, is not conducive to the membranotropic properties of the analyzed glycosides. It should be noted that the influence of sulfate groups on the membranolytic action of the triterpene glycosides depends on the architecture of their carbohydrate chains and the positions of attachment of these functional groups. Hence, increasing the numbers of sulfates in the glycosides with tetrasaccaride and pentasaccharide chains branched by the C-2 of quinovose leads to the activity decreasing [15], which is in good accordance with the observations: trisulfated tetraoside djakonovioside E_1_ (**3**) is weakly hemolytic. So, the complicated and ambiguous character of structure–activity relationships is related to the diverse impact on the membranolytic action of certain structural elements and their combinations in the glycosidic molecules.

Regarding the cytotoxicity of the tested compounds against human breast cancer cells, the selective action of djakonovioside E_1_ (**3**) against the ER-positive MCF-7 cell line and the triple-negative MDA-MB-231 cell line, which fail to express receptors to sex hormones and have no approved targeted therapeutics, was the most important finding, especially amid the absence of a toxic effect in relation to normal mammary epithelial cells (MCF-10A) and low hemolytic activity. The MDA-MB-231 (triple-negative breast cancer) cell line was the most sensitive to cytotoxic action, while the MCF-7 cell line was the most resistant.

The action of djakonoviosides C_1_ (**1**), E_1_ (**3**), and cucumarioside A_2_-5 (**6**), which are the most active against cancer cells, was more deeply studied. It was shown that these compounds did not lose cytotoxicity over time, and djakonovioside E_1_ (**3**) demonstrated an antiproliferative effect. The compounds were able to inhibit colony formation and growth in selected cell lines, with cucumarioside A_2_-5 (**6**) demonstrating the greatest inhibitory effect. The same glycoside was the strongest inhibitor of tumor cell migration. The other glycosides also reliably suppressed cell motility. Such properties of the studied glycosides corroborate their potential to be used as anticancer agents.

## 4. Materials and Methods

### 4.1. General Experimental Procedures

PerkinElmer 343 Polarimeter (PerkinElmer, Waltham, MA, USA) was used for specific rotation measuring; NMR spectra were registered on Bruker AMX 500 (Bruker BioSpin GmbH, Rheinstetten, Germany) (500.12/125.67 MHz (^1^H/^13^C) spectrometer; ESI MS (positive and negative ion modes) spectra were obtained on Agilent 6510 Q-TOF apparatus (Agilent Technology, Santa Clara, CA, USA), sample concentration 0.01 mg/mL; HPLC was conducted on Agilent 1260 Infinity II equipped with a differential refractometer (Agilent Technology, Santa Clara, CA, USA); columns were used: Phenomenex Synergi Fusion RP (10 × 250 mm) and Synergi Hydro RP (10 × 250 mm) (Phenomenex, Torrance, CA, USA) (flow rate 1.5 mL/min), as well as chiral analytical column Kromasil 3-Cellucoat RP (4.6 × 150 mm) (Nouryon HQ, Amsterdam, the Netherlands) (flow rate of 0.5 mL/min).

### 4.2. Animals and Cells

The specimens of sea cucumber *Cucumaria djakonovi* (family Cucumariidae; order Dendrochirotida) were gathered by scuba diving from a depth of 14–15 m near Starichkov’s Island (Avacha Gulf) in July 2007. The taxonomic identification of the animals was performed by Stepanov V.G. Voucher specimen is kept in the Pacific Institute of Geography, Kamchatka Branch, Petropavlovsk-Kamchatsky, Russia.

Human erythrocytes were purchased from the Station of Blood Transfusion, Vladivostok. Human mammary epithelial cell line MCF-10A CRL-10317, human breast cancer cell lines T-47D HTB-133, MCF-7 HTB-22, and MDA-MB-231 CRM-HTB-26 were received from ATCC (Manassas, VA, USA). Culturing conditions: medium of RPMI-1640 with 1% penicillin/streptomycin (Biolot, St. Petersburg, Russia) and 10% fetal bovine serum (FBS) (Biolot, St. Petersburg, Russia) for T-47D cell line; Minimum Essential Medium (MEM) with 1% penicillin/streptomycin sulfate (Biolot, St. Petersburg, Russia) and FBS (Biolot, St. Petersburg, Russia) to a final concentration of 10% for MCF-7 and MDA-MB-231 cells; DMEM/F12 medium with 10% FBS, 20 ng/mL EGF, 0.5 mg/mL hydrocortisone, 100 ng/mL cholera toxin, 10 μg/mL insulin, and 1% penicillin/streptomycin (Bioinnlabs, Russia) for MCF-10A cell line.

### 4.3. Extraction and Isolation

The raw material of the sea cucumbers (663.5 g) was obtained after twice extracting with refluxing 70% EtOH. The extract, dissolved in H_2_O, was chromatographed on a Polychrom-1 column (powdered Teflon, Biolar, Latvia) for the elimination of inorganic salts and impurities. Crude glycoside fraction (1379 mg) was obtained as result of elution with 55% acetone. Its separation by chromatography on Si gel columns (CC) with the stepped gradient of the system of eluents of CHCl_3_/EtOH/H_2_O in ratios of 100:50:4, 100:75:10, 100:100:17, and 100:125:25 gave five fractions. The fractions III, IV, and V were subjected to additional stage of CC with the system of eluents of CHCl_3_/EtOH/H_2_O (100:75:10), (100:100:17), and (100:125:25) that led to isolation of subfractions 3 (262 mg), 4 (1154 mg), and 5 (820 mg), respectively. HPLC of subfraction 3 on reversed-phase column Synergi Fusion RP (10 × 250 mm) with MeOH/H_2_O/NH_4_OAc (1M water solution) in ratio of (68/30/2) as mobile phase gave fractions 3(1)–3(8). The re-chromatography of fractions 3(8) with CH_3_CN/H_2_O/NH_4_OAc (1M water solution) (40/58/2) as mobile phase resulted in the isolation of djakonovioside D_1_ (**2**) (3.6 mg, Rt 17.5 min); 3(7) and 3(5) with CH_3_CN/H_2_O/NH_4_OAc (1M water solution) (39/59/2) gave djakonovioside C_1_ (**1**) (8 mg, Rt 21.2 min), okhotoside A_2_-1 (**5**) (5.6 mg, Rt 18.7 min) from 3(7), and cucumarioside A_2_-5 (**6**) (7 mg, Rt 17.2 min) from 3(5); 3(4) with CH_3_CN/H_2_O/NH_4_OAc (1M water solution) (34/64/2) resulted in the isolation of frondoside A_2_-3 (**7**) (3.3 mg, Rt 15.1 min). Subfraction 4 was subjected to HPLC on the same column with CH_3_CN/H_2_O/NH_4_OAc (1M water solution) (34/64/2) as mobile phase and was separated to two main fractions. One of them was subsequently submitted to HPLC on the Kromasil 3-Cellucoat RP (4.6 × 150 mm) column with CH_3_CN/H_2_O/NH_4_OAc (1M water solution) (20/78/2) as mobile phase to give individual djakonovioside E_1_ (**3**) (5 mg, Rt 11.9 min). From another fraction, cucumarioside A_3_-2 (**8**) (3.1 mg, Rt 15.3 min) was isolated as result of HPLC on Synergi Hydro RP column (10 × 250 mm) with MeOH/H_2_O/NH_4_OAc (1M water solution) in ratio of (67/29/4). The separation of subfraction 5 on Synergi Hydro RP column (10 × 250 mm) with MeOH/H_2_O/NH_4_OAc (1M water solution) in ratio of (66/30/4) gave isokoreoside A (**9**) (3.4 mg, Rt 13.0 min), koreoside A (**10**) (3.4 mg, Rt 12.1 min), and djakonovioside F_1_ (**4**) (4.0 mg, Rt 18.4 min).

#### 4.3.1. Djakonovioside C_1_ (**1**)

Colorless powder; [α]_D_^20^*−*45° (*c* 0.1, H_2_O), mp 215 °C. NMR: Table 1 and Table 2, Appendix A. (*−*)HR-ESI-MS *m*/*z*: 1325.5451 (calc. 1325.5478) [M_Na_ − Na]^−^; (*−*)ESI-MS/MS *m*/*z*: 1265.5 [M_Na_ − Na − CH_3_COOH]^−^, 1223.5 [M_Na_ − Na − NaSO_3_ + H]^−^, 1193.5 [M_Na_ − Na − Xyl (C_5_H_8_O_4_)]^−^, 987.4 [M_Na_ − Na − MeGlc (C_7_H_12_O_5_) − Glc (C_6_H_10_O_5_) + H]^−^, 813.2 [M_Na_ − Na − Agl (C_32_H_47_O_5_) − H]^−^, 681.1 [M_Na_ − Na − Agl (C_32_H_47_O_5_)−Xyl (C_5_H_9_O_4_)]^−^, 595.2 [M_Na_ − Na − Agl (C_32_H_47_O_5_) − XylSO_3_ (C_5_H_7_O_6_SNa)]^−^; (*+*)ESI-MS/MS *m*/*z*: 1251.6 [M_Na_ + Na − NaHSO_4_]^+^, 1179.6 [M_Na_ + Na − MeGlc (C_7_H_13_O_6_) + H]^+^.

#### 4.3.2. Djakonovioside D_1_ (**2**)

Colorless powder; [α]_D_^20^*−*53° (*c* 0.1, H_2_O), mp 219 °C. NMR: Table 3 and Table 4, Appendix A. (*−*)HR-ESI-MS *m*/*z*: 1355.5596 (calc. 1355.5584) [M_Na_ − Na]^−^, 677.2767 (calc. 677.2755) [M_Na_ − Na − H]^2−^; (*−*)ESI-MS/MS *m*/*z*: 1296.5 [M_Na_ − Na − CH_3_COOH]^−^, 1105.5 [M_Na_ − Na − SO_3_Na – Xyl (C_5_H_9_O_5_) + H]^−^, 843.2 [M_Na_ − Na − Agl (C_32_H_47_O_5_) − H]^−^, 723.3 [M_Na_ − Na − Agl (C_32_H_47_O_5_) − NaSO_4_]^−^, 589.2 [M_Na_ − Na − MeGlc (C_7_H_13_O_5_)]^2−^; (*+*)ESI-MS/MS *m*/*z*: 1281.6 [M_Na_ + Na − NaHSO_4_]^+^, 1209.6 [M_Na_ + Na − MeGlc (C_7_H_13_O_6_) + H]^+^.

#### 4.3.3. Djakonovioside E_1_ (**3**)

Colorless powder; [α]_D_^20^ − 50° (*c* 0.1, H_2_O), mp 204 °C. NMR: Table 5 and Appendix A. (−)HR-ESI-MS *m*/*z*: 1427.3942 (calc. 1427.3936) [M_3Na_ − Na]^−^; 702.2037 (calc. 702.2022) [M_3Na_− 2Na]^2−^; 460.4732 (calc. 460.4717) [M_3Na_ − 3Na]^3−^; (*−*)ESI-MS/MS *m*/*z*: 1367.4 [M_3Na_ − Na − CH_3_COOH]^−^, 1307.4 [M_3Na_ − Na − NaHSO_4_]^−^, 1029.4 [M_3Na_ − Na − NaHSO_4_–MeGlcSO_3_ (C_7_H_12_O_9_SNa) + H]^−^, 915.1 [M_3Na_ − Na − Agl (C_32_H_47_O_5_) − H]^−^, 681.1 [M_3Na_ − Na − Agl (C_32_H_47_O_5_) − XylSO_3_ (C_5_H_7_O_7_SNa) − H]^−^, 519.0 [M_3Na_ − Na − Agl (C_32_H_47_O_5_) − XylSO_3_ (C_5_H_7_O_7_SNa) − Glc (C_6_H_10_O_5_) − H]^−^, 446.0 [M_3Na_ − 2Na − Agl (C_32_H_47_O_5_) − H]^2−^.

#### 4.3.4. Djakonovioside F_1_ (**4**)

Colorless powder; [α]_D_^20^*−*32° (*c* 0.1, H_2_O), mp 196 °C. NMR: Table 6 and Table 7, Appendix A. (*−*)HR-ESI-MS *m*/*z*: 1487.4467 (calc. 1487.4511) [M_3Na_ − Na]^−^, 732.2320 (calc. 732.2310) [M_3Na_− 2Na]^2−^, 480.4926 (calc. 480.4909) [M_3Na_ − 3Na]^3−^; (*−*)ESI-MS/MS *m*/*z*: 1489.5 [M_3Na_ − Na + 2]^−^, 1211.4 [M_3Na_ − Na + 2 − MeGlcSO_3_ (C_7_H_11_O_8_SNa)]^−^, 947.4 [M_3Na_ − Na + 2 − MeGlcSO_3_ (C_7_H_11_O_8_SNa) − Glc SO_3_ (C_6_H_9_O_8_SNa)]^−^, 797.1 [M_3Na_ − Na + 2 − MeGlcSO_3_ (C_7_H_11_O_8_SNa) − Glc SO_3_ (C_6_H_9_O_8_SNa) − XylSO_3_ (C_5_H_9_O_5_) − H]^−^.

### 4.4. Cytotoxic Activity (MTT Assay)

The concentrations of tested glycosides were 0.1–50 µM; positive controls—cisplatin and djakonovioside A_1_ [5]. Methodology: to each well of 96-well plates, the cell suspension (180 µL) with solution (20 µL) of tested glycoside in the certain concentration was placed (MCF-10A, MCF-7, T-47D, and MDA-MB-231—7 × 103 cells per well) and incubated in the atmosphere with 5% CO_2_ at 37 °C for 24 h. The times of incubation of cucumarioside A_2_-5 (**6**) and djakonoviosides C_1_ (**1**) and E_1_ (**3**), at concentrations of 0.5–12.0 μM with T-47D, MCF-7, or MDA-MB-231 cells, were 24, 48, and 72 hrs. Then, the solutions of tested compounds with medium were replaced by 100 µL of fresh medium, and 10 µL of 3-(4,5-dimethylthiazol-2-yl)-2,5-diphenyltetrazolium bromide (MTT) (PanReac, AppliChem, Darmstadt, Germany) stock solution (5 mg/mL) was added to each well and incubated for 4 h, followed by the addition of 100 µL of SDS-HCl solution (1 g SDS/10 mL d-H2O/17 µL 6 N HCl) and further incubated for 18 h. Multiskan FC microplate photometer (Thermo Fisher Scientific, Waltham, MA, USA) was used to measure the absorbance of the converted dye formazan at 570 nm. The concentration caused 50% cell metabolic activity inhibition (IC50), which expresses the cytotoxic activity of each glycoside. The experiments were conducted in triplicate, *p* ≤ 0.05.

### 4.5. Hemolytic Activity

Human blood (B(III) Rh+) was used to obtain erythrocytes by centrifuging 450× *g* three times for 5 min with phosphate-buffered saline (PBS) (pH 7.4) at 4 °C on centrifuge LABOFUGE 400R (Heraeus, Hanau, Germany). Ice-cold PBS (pH 7.4) was used for resuspension of erythrocytes residue to a final optical density of 1.5 at 700 nm, which was kept on ice. Then, 20 µL of tested compound solution or control—djakonovioside A_1_ [5]—were added to 180 µL of erythrocyte suspension in V-bottom 96-well plates and exposed for 1 h at 37 °C. Next, centrifugation at 900× *g* for 10 min on laboratory centrifuge LMC-3000 (Biosan, Riga, Latvia), led to layers’ separation, and 100 µL of supernatant was carefully decanted and transferred into new flat-plate each. The values of erythrocyte lysis were measured on microplate photometer Multiskan FC (Thermo Fisher Scientific, Waltham, MA, USA) at λ = 570 nm as hemoglobin concentration in supernatant. The effective dose, causing lysis of 50% erythrocytes (ED50), was calculated with SigmaPlot 14.0 software. All the experiments were carried out in triple repetitions, *p* ≤ 0.05.

### 4.6. Colony Formation Assay

The influence of glycosides on colony formation by MCF-7 or MDA-MB-231 cells was tested by the clonogenic assay [49]. Cell density: 1 × 10^3^ for MDA-MB-231 and 0.3 × 10^3^ for MCF-7 cells per well; cell culture: MEM media, 10% FBS, 10,000 U/mL of penicillin, and 10,000 μg/mL of streptomycin supplemented or not (control) with different concentration of glycosides; incubation conditions: 10 days, 37 °C, atmosphere with 5% CO_2_; obtained: visible to eye colonies (at least 50 cells per colony); fixation with methanol (25 min); staining with 0.5% solution of crystal violet (25 min); washing and air-drying of plates.

### 4.7. Wound Scratch Migration Assay

Attached to special migration plate plastic bottom MCF-7 or MDA-MB-231 cells were separated by a silicone insert (Culture-insert 2 Well 24, ibiTreat). After removing an insert, gap between the cells was 500 ± 50 μm. Cell debris and floating cells were deleted by washing twice with PBS; 10 μM of 5 mM initial solution of CFDA SE ((5,6)-carboxyfluorescein succinimidyl ester) (LumiTrace CFDA SE kit, Lumiprobe, Moscow, Russia) in DMSO was dissolved in PBS and added to cells for 5 min at 37 °C; after washing twice with PBS, the fresh culture medium was added. Then, cells were treated with various concentrations of glycosides or culture medium only (vehicle control) and left for 24 and 72 h. Cell migration into the wound area was observed under a fluorescence microscope (MIB-2-FL, LOMO, Russia) with objective 10× magnification. 

### 4.8. Building a QSAR Model

QSAR model for the set of 20 glycosides was built using QuaSAR-Descriptor and QuaSAR-Model tools of MOE 2020.0901 software [22]. The procedure involved the following steps: a charge calculation and structure optimization, glycosides conformational search, descriptors calculation, correlational analysis, principal component analysis (PCA), removing the descriptors collinear with another descriptor (unnecessary descriptors), building a QSAR model and the model’s cross-validation, removing the descriptors not contributing to the model, and model checking by making a graph showing the correlation between the model-predicted value and the experimental activity value expressed as pED_50_.

## 5. Conclusions

As result of thorough research on the glycosidic composition of the sea cucumber *Cucumaria djakonovi*, 11 new djakonoviosides and 9 known glycosides, found earlier in other representatives of the *Cucumaria* genus, were isolated in general. Twelve different aglycones were the parts of the found compounds, and five of them were new ones, including a unique one having a 23,16-hemiketal fragment. Nine types of carbohydrate chains composed the glycosides of *C. djakonovi*. Two types of sugar moieties were revealed first, including those with xylose or glucose residue in the second position of the chain.

The pathways of biosynthesis of the aglycones and carbohydrate moieties of *C. djakonovi*’s glycosides were proposed, showing the regularities characteristic of the mosaic type of biosynthesis.

It was shown that, on the one hand, the unique species-specific composition of the glycosides is inherent for each species of the genus *Cucumaria*, and, on the other hand, the presence of some common glycosides for all representatives of the genus is typical.

Some of the glycosides from *C. djakonovi* display promising anti-breast-cancer effects expressed as the inhibition of cells’ viability, functioning, and motility—the aspects of carcinogenesis occurring in the organism.

Quantitative structure–activity relationships analysis confirmed the complicated, tricky character of these correlations because of the influence on the activity of a large number of properties and peculiarities of the molecules that act in combination altogether. Importantly, the calculated results of QSAR are in good accordance with the observed SAR, concluded on the basis of the experimental data. All these indicate that the application of this instrument for the prediction or modeling of the biologic activity of sea cucumbers’ triterpene glycosides is rather prospective and useful.

## Figures and Tables

**Figure 1 marinedrugs-21-00602-f001:**
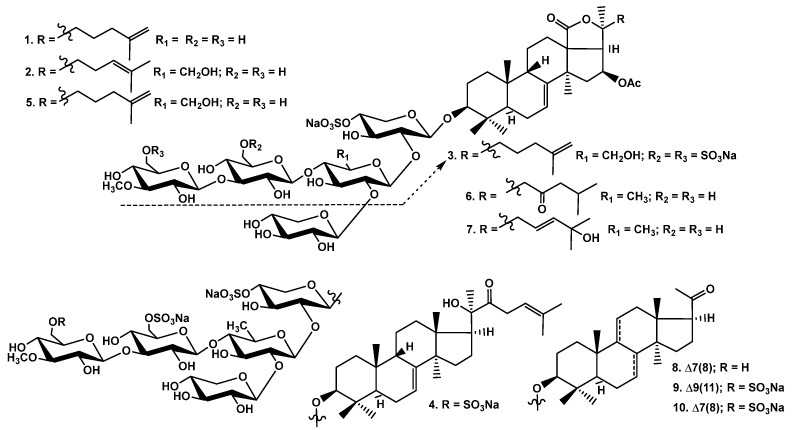
Chemical structures of glycosides of *Cucumaria djakonovi*: **1**—djakonovioside C_1_; **2**—djakonovioside D_1_; **3**—djakonovioside E_1_; **4**—djakonovioside F_1_; **5**—okhotoside A_2_-1; **6**—cucumarioside A_2_-5; **7**—frondoside A_2_-3; **8**—cucumarioside A_3_-2; **9**—isokoreoside A; **10**—koreoside A.

**Figure 2 marinedrugs-21-00602-f002:**
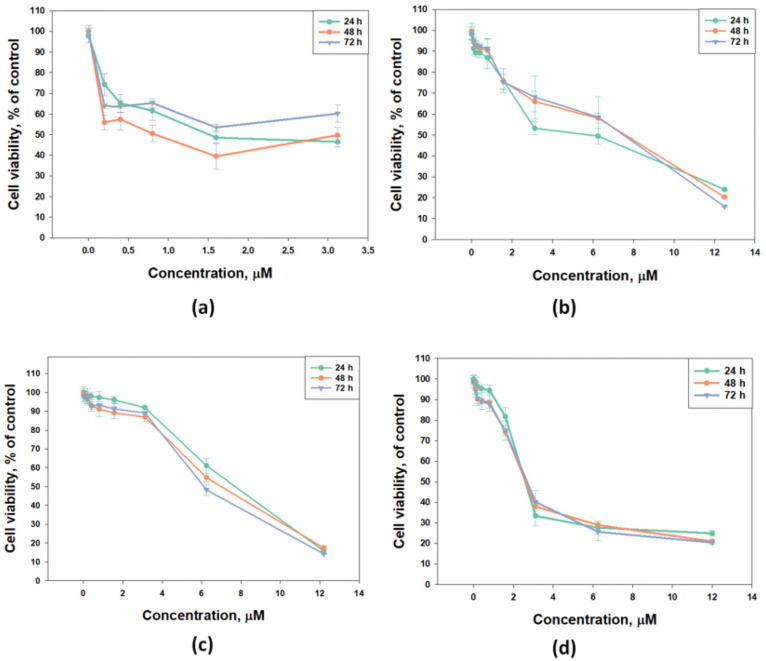
Cytotoxic effect of glycosides against breast cancer cells: (**a**) djakonovioside E_1_ (**3**) against MCF-7, (**b**) cucumarioside A_2_-5 (**6**) against T-47D cells, (**c**) djakonovioside C_1_ (**1**) against MDA-MB-231 cells, and (**d**) cucumarioside A_2_-5 (**6**) against MDA-MB-231 cells for 24 h, 48 h, and 72 h. All experiments were carried out in triplicate. The data are presented as mean ± SEM.

**Figure 3 marinedrugs-21-00602-f003:**
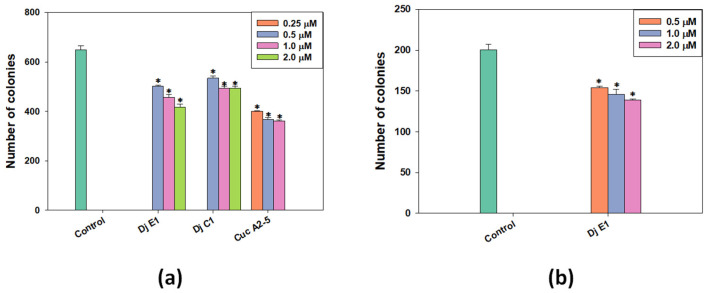
The number of MDA-MB-231 (**a**) and MCF-7 (**b**) cell colonies under the treatment with different concentrations of djakonovioside C_1_ (**1**), djakonovioside E_1_ (**3**), and cucumarioside A_2_-5 (**6**). ImageJ 1.52 software was used to count the cell colonies. Data are presented as means ± SEM. * *p* value ≤ 0.05 considered significant.

**Figure 4 marinedrugs-21-00602-f004:**
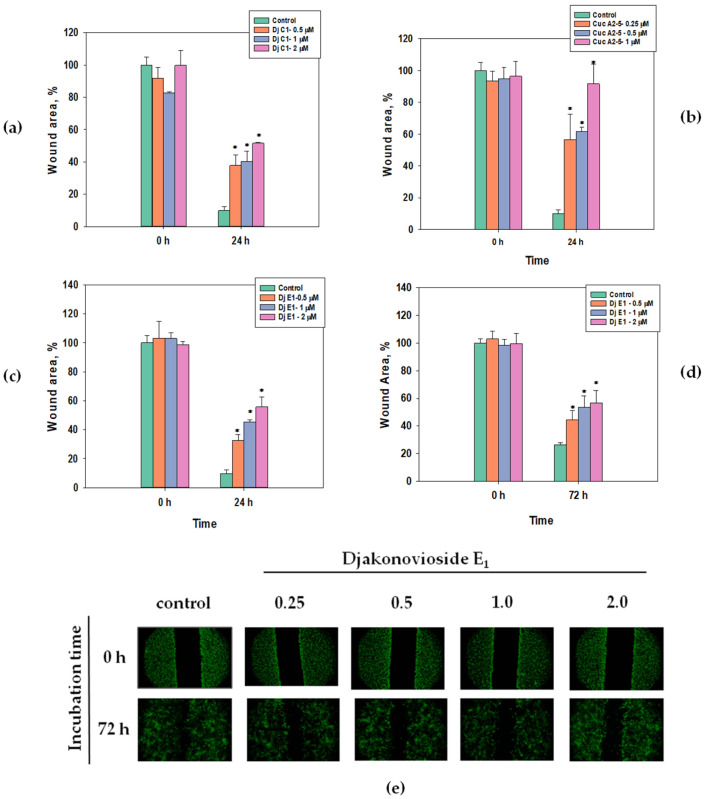
Migration of MDA-MB-231 and MCF-7 cells into wound areas observed with MIB-2-FL fluorescence microscope at 10-fold magnification: (**a**) MDA-MB-231 cells 0 and 24 h after treatment with 0.5, 1.0, and 2.0 μM of djakonovioside C_1_ (**1**); (**b**) MDA-MB-231 cells 0 and 24 h after treatment with 0.25, 0.5, and 1.0 μM of cucumarioside A_2_-5 (**6**); (**c**) MDA-MB-231 cells 0 and 24 h after treatment with 0.5, 1.0, and 2.0 μM of djakonovioside E_1_ (**3**); (**d**,**e**) MCF-7 cells 0 and 72 h after treatment with 0.5, 1.0, and 2.0 μM of djakonovioside E_1_ (**3**). Cells were stained with the fluorescent dye CFDA SE. Cell migration into wound areas processed by ImageJ 1.52 software. Data are presented as means ± SEM. * *p* value ≤ 0.05 considered significant.

**Figure 5 marinedrugs-21-00602-f005:**
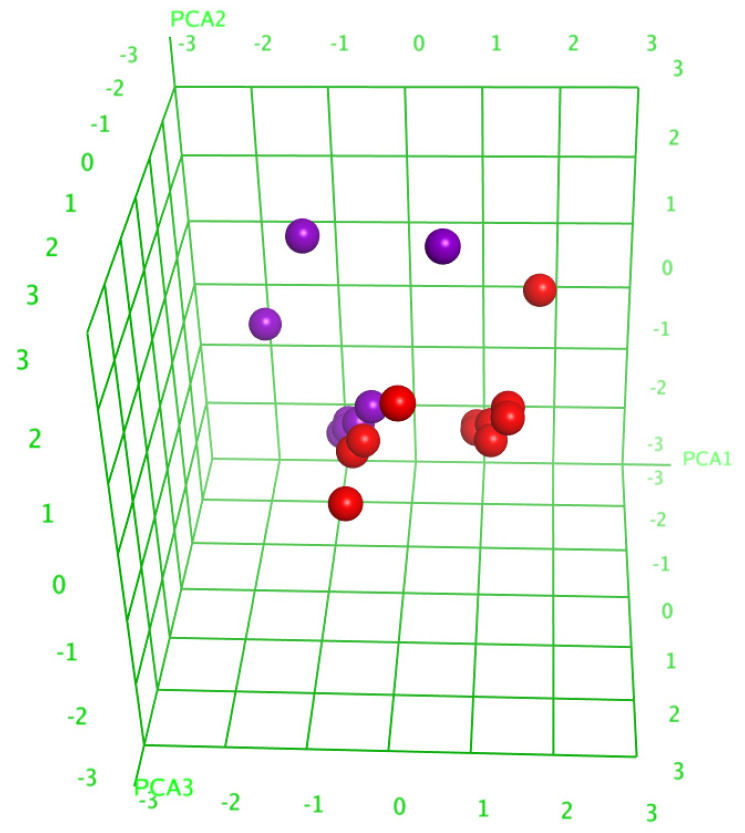
Three-dimensional plot of hemolytic activity (pED_50_) depending on the principal components’ values (PCA1—PCA3) calculated for 20 glycosides. The glycosides that demonstrated hemolytic activity with ED_50_ ≤ 10 µM were outlined as active and are marked in red, while the rest are marked in violet.

**Figure 6 marinedrugs-21-00602-f006:**
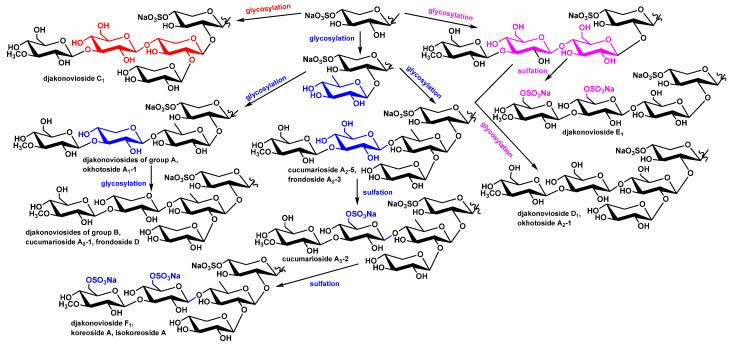
The scheme of biosynthesis of carbohydrate chains of the glycosides of *C. djakonovi*.

**Table 1 marinedrugs-21-00602-t001:** ^13^C and ^1^H NMR chemical shifts and HMBC and ROESY correlations of the aglycone moiety of djakonovioside C_1_ (**1**).

Position	δ_C_ mult. ^a^	δ_H_ mult. (*J* in Hz) ^b^	HMBC	ROESY
1	35.9 CH_2_	1.31 m		H-3, H-11, H-19
2	26.8 CH_2_	1.95 m		
		1.78 m		H-19, H-30
3	88.8 CH	3.20 dd (4.7; 12.1)		H-1, H-5, H-31, H1-Xyl1
4	39.3 C			
5	47.9 CH	0.91 dd (4.7; 10.7)	C: 19	H-1, H-3, H-31
6	23.1 CH_2_	1.91 m		H-19, H-30, H-31
7	120.2 CH	5.57 m		H-15, H-32
8	145.5 C			
9	47.0 CH	3.30 brd (14.1)		H-19
10	35.3 C			
11	22.4 CH_2_	1.72 m		H-1
		1.47 m		H-32
12	31.2 CH_2_	2.10 m		H-17, H-21, H-32
13	59.3 C			
14	47.3 C			
15	43.5 CH_2_	2.54 dd (7.4; 12.1)	C: 13, 17, 32	H-7, H-32
		1.61 dd (8.7; 12.1)		
16	75.3 CH	5.82 dd (8.7; 16.1)		H-32
17	54.5 CH	2.66 d (8.7)	C: 12, 13, 18, 21	H-12, H-16, H-21, H-32
18	180.2 C			
19	23.8 CH_3_	1.06 s	C: 5, 9, 10	H-1, H-2, H-6, H-9
20	85.5 C			
21	28.0 CH_3_	1.51 s	C: 17, 20, 22	H-12, H-17, H-22
22	38.2 CH	2.25 td (4.7; 12.7)		
		1.80 m		H-21
23	22.9 CH_2_	1.47 m		
		1.35 m		
24	38.1 CH_2_	1.91 m		
25	145.4 C			
26	110.8 CH_2_	4.72 m	C: 24, 27	H-24
27	22.0 CH_3_	1.65 s	C: 24, 25, 26	
30	17.1 CH_3_	1.01 s	C: 3, 4, 5, 31	H-2, H-6, H-31
31	28.4 CH_3_	1.17 s	C: 3, 4, 5, 30	H-3, H-5, H-6, H-30, H-1 Xyl1
32	32.1 CH_3_	1.15 s	C: 8, 13, 14, 15	H-7, H-12, H-15, H-16, H-17
O**C**OCH_3_	170.7 C			
OCO**C**H_3_	21.2 CH_3_	2.01 s	OAc	OAc

^a^ Recorded at 125.67 MHz in C_5_D_5_N/D_2_O (4/1). ^b^ Recorded at 500.12 MHz in C_5_D_5_N/D_2_O (4/1). The original spectra of **1** are provided in Appendix A.

**Table 2 marinedrugs-21-00602-t002:** ^13^C and ^1^H NMR chemical shifts and HMBC and ROESY correlations of carbohydrate moiety of djakonovioside C_1_ (**1**).

Atom	δ_C_ mult. ^a^	δ_H_ mult. (*J* in Hz) ^b,c,d^	HMBC	ROESY
Xyl1 (1→C-3)				
1	104.7 CH	4.73 d (7.5)	C: 3	H-3; H-3, 5 Xyl1
2	**81.2** CH	3.97 t (8.8)	C: 1 Xyl2; C: 1 Xyl1	H-1 Xyl2
3	75.3 CH	4.30 t (8.8)	C: 2, 4 Xyl1	H-1 Xyl1
4	*76.1* CH	5.02 m		H-2 Xyl1
5	64.2 CH_2_	4.80 dd (4.8; 11.0)	C: 1, 3 Xyl1	
		3.85 dd (8.8; 11.0)		H-1 Xyl1
Xyl2 (1→2Xyl1)				
1	102.7 CH	5.25 d (6.8)	C: 2 Xyl1	H-2 Xyl1; H-3, 5 Xyl2
2	**82.6** CH	3.95 dd (6.8; 8.9)	C: 1 Xyl2; C: 1 Xyl5	H-1 Xyl5
3	74.7 CH	4.12 t (8.9)	C: 4 Xyl2	H-1 Xyl2
4	**77.9** CH	4.19 m	C: 1 Glc3	H-1 Glc3
5	63.7 CH_2_	4.39 dd (5.3; 11.3)		
		3.55 dd (9.0; 11.3)		H-1 Xyl2
Glc3 (1→4Xyl2)				
1	102.4 CH	4.85 d (7.5)	C: 4 Xyl2	H-4 Xyl2; H-3, 5 Glc3
2	73.3 CH	3.90 t (9.0)	C: 1 Glc3	
3	**86.8** CH	4.15 t (9.0)	C: 2, 4 Glc3; C: 1 MeGlc4	H-1 MeGlc4; H-1, 5 Glc3
4	69.3 CH	3.88 t (9.0)	C: 3, 5 Glc3	
5	77.2 CH	3.80 m		
6	61.6 CH_2_	4.26 dd (3.0; 11.3)		H-1 Glc3
		4.02 dd (6.0; 11.3)	C: 5 Glc3	
MeGlc4 (1→3Glc3)				
1	104.6 CH	5.19 d (8.0)	C: 3 Glc3	H-3 Glc3; H-3, 5 MeGlc4
2	74.6 CH	3.85 t (8.8)	C: 1, 3 MeGlc4	
3	87.0 CH	3.67 t (8.8)	C: 2, 4 MeGlc4; OMe	H-1 MeGlc4; OMe
4	70.4 CH	3.88 t (8.8)		
5	77.5 CH	3.92 m		H-1 MeGlc4
6	61.7 CH_2_	4.37 dd (3.2; 12.0)		
		4.04 dd (6.4; 12.0)		
OMe	60.7 CH_3_	3.80 s	C: 3 MeGlc4	
Xyl5 (1→2Xyl2)				
1	105.6 CH	5.09 d (7.2)	C: 2 Xyl2	H-2 Xyl2; H-3, 5 Xyl5
2	74.8 CH	3.91 t (8.8)	C: 1, 3 Xyl5	
3	76.6 CH	4.01 t (8.8)	C: 2, 4 Xyl5	H-1 Xyl5
4	70.0 CH	4.07 m		
5	66.5 CH_2_	4.28 dd (5.6; 12.0)	C: 3, 4 Xyl5	
		3.56 t (10.4)	C: 3, 4 Xyl5	H-1 Xyl5

^a^ Recorded at 125.67 MHz in C_5_D_5_N/D_2_O. ^b^ Bold—interglycosidic positions. ^c^ Italics—sulfate position. ^d^ Recorded at 500.12 MHz in C_5_D_5_N/D_2_O. Multiplicity by 1D TOCSY. The original spectra of **1** are provided in Appendix A.

**Table 3 marinedrugs-21-00602-t003:** ^13^C and ^1^H NMR chemical shifts and HMBC and ROESY correlations of aglycone moiety of djakonovioside D_1_ (**2**).

Position	δ_C_ mult. ^a^	δ_H_ mult. (*J* in Hz) ^b^	HMBC	ROESY
1	35.9 CH_2_	1.29 m		H-11
2	26.7 CH_2_	1.93 m		
		1.85 m		
3	89.2 CH	3.21 dd (3.9; 11.6)		H-5, H1-Xyl1
4	39.4 C			
5	47.8 CH	0.89 dd (5.2; 11.3)	C: 6, 10, 19, 30	H-3, H-31
6	23.0 CH_2_	1.88 m		H-30, H-31
7	120.2 CH	5.58 m		H-15
8	145.5 C			
9	47.0 CH	3.29 brd (14.6)		H-19
10	35.3 C			
11	22.4 CH_2_	1.72 m		H-1
		1.48 m		
12	31.2 CH_2_	2.10 m		H-21
13	59.3 C			
14	47.3 C			
15	43.5 CH_2_	2.57 dd (7.5; 12.0)	C: 13, 17, 32	H-7
		1.61 m		
16	75.2 CH	5.85 dd (8.2; 16.7)		H-32
17	54.3 CH	2.70 d (8.2)	C: 12, 13, 18, 21	H-32
18	180.2 C			
19	23.7 CH_3_	1.06 s	C: 5, 9, 10	H-2, H-6, H-9
20	85.5 C			
21	28.0 CH_3_	1.56 s	C: 17, 20, 22	H-12, H-17, H-23
22	38.5 CH	2.42 td (4.1; 12.3)		
		1.83 m		
23	23.5 CH_2_	2.03 m		
		1.91 m		
24	123.9 CH	5.00 m		
25	132.1 C			
26	25.4 CH_3_	1.60 s	C: 24, 25, 27	H-24
27	17.6 CH_3_	1.54 s	C: 24, 25, 26	
30	17.2 CH_3_	1.05 s	C: 3, 4, 5, 31	H-2, H-6, H-31
31	28.5 CH_3_	1.18 s	C: 3, 4, 5, 30	H-3, H-30, H-1 Xyl1
32	32.1 CH_3_	1.17 s	C: 8, 13, 14, 15	H-15, H-16, H-17
O**C**OCH_3_	170.7 C			
OCO**C**H_3_	21.0 CH_3_	1.97 s	OAc	OAc

^a^ Recorded at 125.67 MHz in C_5_D_5_N/D_2_O. ^b^ Recorded at 500.12 MHz in C_5_D_5_N/D_2_O. The original spectra of **2** are provided in Appendix A.

**Table 4 marinedrugs-21-00602-t004:** ^13^C and ^1^H NMR chemical shifts and HMBC and ROESY correlations of carbohydrate moiety of djakonovioside D_1_ (**2**).

Atom	δ_C_ mult. ^a^	δ_H_ mult. (*J* in Hz) ^b,c,d^	HMBC	ROESY
Xyl1 (1→C-3)				
1	104.7 CH	4.73 d (8.5)	C: 3	H-3; H-3, 5 Xyl1
2	**79.7** CH	4.18 t (8.5)	C: 1 Glc2; C: 1 Xyl1	H-4 Xyl1; H-1 Glc2
3	75.5 CH	4.34 t (9.2)	C: 4 Xyl1	H-1 Xyl1
4	*76.1* CH	4.99 m		
5	64.2 CH_2_	4.78 dd (5.0; 11.4)	C: 4 Xyl1	
		3.79 dd (8.5; 11.4)		
Glc2 (1→2Xyl1)				
1	101.4 CH	5.39 d (6.8)	C: 2 Xyl1	H-2 Xyl1; H-5 Glc2
2	**82.2** CH	3.96 t (8.9)	C: 1 Xyl5, C: 1, 3 Glc2	H-1 Xyl5
3	75.5 CH	4.08 t (8.9)	C: 2, 4 Glc2	H-1 Glc2
4	**80.6** CH	4.01 t (8.9)	C: 1 Glc3; C: 5 Glc2	H-1 Glc3
5	75.8 CH	3.68 m		H-1 Glc2
6	61.4 CH_2_	4.34 brd (11.6)		
		4.25 dd (5.5; 11.6)		
Glc3 (1→4Glc2)				
1	103.6 CH	4.96 d (8.0)	C: 4 Glc2	H-4 Glc2; H-3, 5 Glc3
2	73.5 CH	3.91 t (8.6)	C: 1, 3 Glc3	
3	**86.7** CH	4.16 t (8.6)	C: 2, 4 Glc3	H-1 MeGlc4
4	69.2 CH	3.87 m		
5	77.0 CH	3.85 m		H-1 Glc3
6	61.1 CH_2_	4.24 brd (9.6)		
		3.99 dd (4.8; 11.8)		
MeGlc4 (1→3Glc3)				
1	104.5 CH	5.16 d (7.6)	C: 3 Glc3	H-3 Glc3; H-3, 5 MeGlc4
2	74.5 CH	3.85 t (8.3)	C: 1, 3 MeGlc4	
3	86.9 CH	3.67 t (8.3)	OMe	H-1 MeGlc4; OMe
4	70.1 CH	3.89 t (8.3)	C: 5 MeGlc4	
5	77.5 CH	3.91 m		H-1 MeGlc4
6	61.7 CH_2_	4.35 brd (11.9)	C: 4 MeGlc4	
		4.03 dd (5.9; 11.9)	C: 4, 5 MeGlc4	
OMe	60.7 CH_3_	3.81 s	C: 3 MeGlc4	
Xyl5 (1→2Glc2)				
1	105.2 CH	5.19 d (7.1)	C: 2 Glc2	H-2 Glc2; H-3, 5 Xyl5
2	74.7 CH	3.94 t (8.3)	C: 1, 3 Xyl5	
3	76.3 CH	4.04 t (8.3)	C: 2 Xyl5	H-1 Xyl5
4	70.3 CH	4.09 dd (5.1; 9.0)		
5	66.3 CH_2_	4.31 dd (5.1; 11.5)		
		3.59 t (10.3)		H-1 Xyl5

^a^ Recorded at 125.67 MHz in C_5_D_5_N/D_2_O. ^b^ Recorded at 500.12 MHz in C_5_D_5_N/D_2_O. ^c^ Bold—interglycosidic positions. ^d^ Italics—sulfate positions. Multiplicity by 1D TOCSY. The original spectra of **2** are provided in Appendix A.

**Table 5 marinedrugs-21-00602-t005:** ^13^C and ^1^H NMR chemical shifts and HMBC and ROESY correlations of carbohydrate moiety of djakonovioside E_1_ (**3**).

Atom	δ_C_ mult. ^a^	δ_H_ mult. (*J* in Hz) ^b,c,d^	HMBC	ROESY
Xyl1 (1→C-3)				
1	104.7 CH	4.68 d (7.3)	C: 3; C: 5 Xyl1	H-3; H-3, 5 Xyl1
2	**81.2** CH	4.07 dd (9.0; 7.3)	C: 1 Glc2; C: 3 Xyl1	
3	74.6 CH	4.25 t (9.0)	C: 2, 4 Xyl1	H-1 Xyl1
4	*76.0* CH	4.94 ddd (14.3; 9.0; 6.4)	C: 3 Xyl1	
5	63.8 CH_2_	4.74 dd (12.2; 5.3)	C: 3 Xyl1	
		3.72 dd (12.2; 10.1)	C: 1 Xyl1	H-1, 3 Xyl1
Glc2 (1→2Xyl1)				
1	104.3 CH	5.07 d (8.0)	C: 2 Xyl1; C: 2 Glc2	H-2 Xyl1; H-3, 5 Glc2
2	75.1 CH	3.85 t (8.7)	C: 3 Glc2	
3	75.2 CH	3.97 t (8.7)	C: 4 Glc2	H-1 Glc2
4	**81.8** CH	3.95 t (8.7)	C: 1 Glc3; C: 3 Glc2	H-1 Glc3
5	75.9 CH	3.68 t (8.7)	C: 4 Glc2	H-1 Glc2
6	61.1 CH_2_	4.26 m		
Glc3 (1→4Glc2)				
1	103.8 CH	4.87 d (8.5)	C: 4 Glc2	H-4 Glc2; H-3, 5 Glc3
2	73.4 CH	3.81 t (8.7)	C: 1, 3 Glc3	
3	**86.4** CH	4.07 t (8.7)	C: 1 MeGlc4; C: 2, 4 Glc3	H-1 MeGlc4; H-1 Glc3
4	69.4 CH	3.74 t (8.7)	C: 5, 6 Glc3	
5	74.8 CH	4.03 m		
6	*67.4* CH_2_	4.93 brd (10.9)		
		4.55 dd (10.9; 6.5)	C: 5 Glc3	
MeGlc4 (1→3Glc3)				
1	104.7 CH	5.09 d (7.9)	C: 3 Glc3; C: 2 MeGlc4	H-3 Glc3; H-3, 5 MeGlc4
2	74.3 CH	3.76 t (8.5)	C: 1 MeGlc4	
3	86.4 CH	3.61 t (8.5)	OMe; C: 2, 4 MeGlc4	H-1, 5 MeGlc4
4	69.8 CH	3.96 m	C: 3, 5, 6 MeGlc4	
5	75.4 CH	4.00 m	C: 4 MeGlc4	H-1 MeGlc4
6	*67.0* CH_2_	4.91 brd (11.2)	C: 4 MeGlc4	
		4.70 brd (9.3)	C: 4, 5 MeGlc4	
OMe	60.5 CH_3_	3.75 s	C: 3 MeGlc4	

^a^ Recorded at 125.67 MHz in C_5_D_5_N/D_2_O. ^b^ Recorded at 500.12 MHz in C_5_D_5_N/D_2_O. ^c^ Bold—interglycosidic positions. ^d^ Italics—sulfate positions. Multiplicity by 1D TOCSY. The original spectra of **3** are provided in Appendix A.

**Table 6 marinedrugs-21-00602-t006:** ^13^C and ^1^H NMR chemical shifts and HMBC and ROESY correlations of aglycone moiety of djakonovioside F_1_ (**4**).

Position	δ_C_ mult. ^a^	δ_H_ mult. (*J* in Hz) ^b^	HMBC	ROESY
1	35.6 CH_2_	1.30 m		H-3, H-5
2	26.9 CH_2_	1.94 m		H-31,
		1.77 m		H-19, H-30
3	88.8 CH	3.14 dd (3.8; 11.4)	C: 30, 31, C: 1 Xyl1	H-1, H-5, H-31, H1-Xyl1
4	39.4 C			
5	49.6 CH	0.85 dd (3.2; 11.4)	C: 4, 10, 19, 30	H-1, H-3, H-31
6	23.1 CH_2_	1.93 m		H-19
		1.85 m		
7	122.1 CH	5.61 m	C: 9, 13	H-15, H-32
8	148.5 C			
9	48.1 CH	2.29 brd (13.3)		H-18, H-19
10	35.5 C			
11	22.8 CH_2_	1.66 m		
		1.42 m		
12	34.9 CH_2_	1.97 m		H-17, H-21
		1.76 m		
13	52.8 C			
14	45.1 C			
15	33.4 CH_2_	1.64 m		H-18
		1.56 m	C: 14, 32	H-7
16	22.3 CH_2_	2.02 m		H-18
		1.56 m		H-21, H-32
17	53.2 CH	2.38 t (8.9)	C: 14, 16, 18, 21	H-12, H-21, H-32
18	24.5 CH_3_	1.29 s	C: 12, 13, 14	H-9, H-12, H-15, H-16, H-19
19	24.6 CH_3_	0.93 s	C: 1, 5, 9, 10	H-2, H-6, H-9, H-18, H-30
20	81.4 C			
21	24.9 CH_3_	1.60 s	C: 17, 20, 22	H-12, H-17, H-18
22	216.1 C			
23	37.0 CH_2_ *	3.61 m		
24	117.3 CH	5.46 m	C: 26, 27	H-26
25	134.9 C			
26	25.5 CH_3_	1.63 s	C: 24, 25, 27	H-24
27	17.9 CH_3_	1.56 s	C: 24, 25, 26	
30	17.5 CH_3_	1.03 s	C: 3, 4, 5, 31	H-2, H-6, H-19, H-31
31	28.7 CH_3_	1.18 s	C: 3, 4, 5, 30	H-3, H-5, H-6, H-30, H-1 Xyl1
32	30.7 CH_3_	1.07 s	C: 8, 13, 14, 15	H-7, H-12, H-15, H-17

^a^ Recorded at 125.67 MHz in C_5_D_5_N/D_2_O. ^b^ Recorded at 500.12 MHz in C_5_D_5_N/D_2_O. * Recorded at 176.04 MHz in C_5_D_5_N/H_2_O. The original spectra of **4** are provided in Appendix A.

**Table 7 marinedrugs-21-00602-t007:** ^13^C and ^1^H NMR chemical shifts and HMBC and ROESY correlations of carbohydrate moiety of djakonovioside F_1_ (**4**).

Atom	δ_C_ mult. ^a^	δ_H_ mult. (*J* in Hz) ^b,c,d^	HMBC	ROESY
Xyl1 (1→C-3)				
1	104.6 CH	4.71 d (7.4)	C: 3	H-3; H-3, 5 Xyl1
2	**81.5** CH	3.97 dd (9.1; 7.4)	C: 1 Qui2; C: 1, 3 Xyl1	H-1 Qui2; H-4 Xyl1
3	75.2 CH	4.30 t (9.1)	C: 2, 4 Xyl1	H-1, 5 Xyl1
4	*76.1* CH	5.00 ddd (13.8; 9.1; 5.6)	C: 3 Xyl1	H-2 Xyl1
5	64.1 CH_2_	4.79 dd (12.6; 5.6)	C: 1, 3 Xyl1	
		3.86 t (10.8)	C; 1, 4 Xyl1	H-1, 3 Xyl1
Qui2 (1→2Xyl1)				
1	102.1 CH	5.18 d (7.2)	C: 2 Xyl1	H-2 Xyl1; H-5 Qui2
2	82.45CH	3.91 t (8.6)	C: 1, 3 Qui2; C: 1 Xyl5	H-1 Xyl5; H-4 Qui2
3	75.2 CH	3.97 t (8.6)	C: 2, 4 Qui2	H-1, 5 Qui2
4	**86.4** CH	3.43 t (8.6)	C: 1 Glc3; C: 3, 5 Qui2	H-1 Glc3; H-2, 6 Qui2
5	70.8 CH	3.56 dd (8.6; 5.9)	C: 4 Qui2	H-1, 3 Qui2
6	17.9 CH_3_	1.56 d (5.9)	C: 5, 6 Qui2	H-4 Qui2
Glc3 (1→4Qui2)				
1	103.9 CH	4.76 d (8.1)	C: 4 Qui2	H-4 Qui2; H-3, 5 Glc3
2	73.4 CH	3.80 t (8.8)	C: 1, 3 Glc3	
3	**86.6** CH	4.10 t (8.8)	C: 1 MeGlc4; C: 2, 4 Glc3	H-1 MeGlc4; H-1 Glc3
4	69.1 CH	3.80 m	C: 3, 5, 6 Glc3	
5	74.9 CH	4.06 m		
6	*67.3* CH_2_	4.94 brd (10.8)		
		4.58 dd (10.8; 5.4)	C: 5 Glc3	
MeGlc4 (1→3Glc3)				
1	104.8 CH	5.13 d (8.1)	C: 3 Glc3; C: 2 MeGlc4	H-3 Glc3; H-3, 5 MeGlc4
2	74.4 CH	3.76 t (8.8)	C: 1 MeGlc4	H-4 MeGlc4
3	86.4 CH	3.62 t (8.8)	OMe; C: 4 MeGlc4	H-1 MeGlc4
4	69.8 CH	3.99 m	C: 5 MeGlc4	
5	75.6 CH	3.99 m		H-1, 3 MeGlc4
6	*67.0* CH_2_	4.93 d (11.4)	C: 4, 5 MeGlc4	
		4.75 brd (11.4)	C: 5 MeGlc4	
OMe	60.5 CH_3_	3.75 s	C: 3 MeGlc4	
Xyl5 (1→2Qui2)				
1	105.2 CH	5.20 d (7.4)	C: 2 Qui2	H-2 Qui2; H-3,5 Xyl5
2	74.9 CH	3.92 t (8.1)	C: 3 Xyl5	
3	76.3 CH	4.07 t (8.1)	C: 2, 4 Xyl5	H-1 Xyl5
4	70.1 CH	4.04 m		
5	66.4 CH_2_	4.27 dd (11.8; 5.2)	C: 1, 3, 4 Xyl5	H-1, 3 Xyl5
		3.63 t (10.3)		

^a^ Recorded at 125.67 MHz in C_5_D_5_N/D_2_O. ^b^ Recorded at 500.12 MHz in C_5_D_5_N/D_2_O. ^c^ Bold—interglycosidic positions. ^d^ Italics—sulfate positions. Multiplicity by 1D TOCSY. The original spectra of **4** are provided in Appendix A.

**Table 8 marinedrugs-21-00602-t008:** ^13^C and ^1^H NMR chemical shifts and HMBC and ROESY correlations of aglycone moiety of cucumarioside A_3_-2 (**8**).

Position	δ_C_ mult. ^a^	δ_H_ mult. (*J* in Hz) ^b^	HMBC	ROESY
1	35.5 CH_2_	1.29 m		H-3, H-11, H-19
2	26.8 CH_2_	1.93 m		H-31
		1.74 m		H-19, H-30
3	88.8 CH	3.15 dd (4.1; 11.6)	C: 4, 30, 31, C: 1 Xyl1	H-1, H-5, H-31, H1-Xyl1
4	39.4 C			
5	48.7 CH	0.85 dd (3.3; 11.6)	C: 4, 6, 10, 19, 30	H-3, H-31
6	23.1 CH_2_	1.90 m		
		1.81 m		H-19, H-30
7	122.5 CH	5.58 m	C: 9, 14	H-15
8	147.6 C			
9	48.1 CH	2.13 m	C: 11	H-18, H-19
10	35.5 C			
11	22.3 CH_2_	1.68 m		H-1, H-18
		1.40 m		H-32
12	33.3 CH_2_	2.00 m		H-17
		1.67 m		H-18
13	44.9 C			
14	53.1 C			
15	33.3 CH_2_	1.68 m		
		1.52 m		H-7, H-18
16	22.4 CH_2_	2.28 m		
17	61.9 CH	2.75 t (8.7)	C: 13, 18, 20	H-12, H-15, H-21, H-32
18	24.7 CH_3_	0.78 s	C: 12, 13, 14, 17	H-9, H-12, H-15, H-16
19	24.3 CH_3_	0.91 s	C: 1, 9, 10	H-1, H-2, H-6, H-9, H-30
20	211.6 C			
21	30.6 CH_3_	2.16 s	C: 17, 20	H-12, H-17
30	17.3 CH_3_	1.02 s	C: 3, 4, 5, 31	H-2, H-6, H-19, H-31
31	28.7 CH_3_	1.18 s	C: 3, 4, 5, 30	H-3, H-5, H-6, H-30, H-1 Xyl1
32	30.4 CH_3_	1.04 s	C: 8, 13, 14, 15	H-7, H-11, H-12, H-15, H-17

^a^ Recorded at 125.67 MHz in C_5_D_5_N/D_2_O. ^b^ Recorded at 500.12 MHz in C_5_D_5_N/D_2_O. The original spectra of **8** are provided in Appendix A.

**Table 9 marinedrugs-21-00602-t009:** ^13^C and ^1^H NMR chemical shifts and HMBC and ROESY correlations of carbohydrate moiety of cucumarioside A_3_-2 (**8**).

Atom	δ_C_ mult. ^a^	δ_H_ mult. (*J* in Hz) ^b,c,d^	HMBC	ROESY
Xyl1 (1→C-3)				
1	104.5 CH	4.72 d (7.0)	C: 3; C: 5 Xyl1	H-3; H-3, 5 Xyl1
2	**81.4** CH	3.96 t (8.3)	C: 1 Qui2; C: 1, 3 Xyl1	H-4 Xyl1; H-1 Qui2
3	75.0 CH	4.31 t (8.3)	C: 2, 4 Xyl1	
4	*76.1* CH	5.00 m	C: 3 Xyl1	H-2 Xyl1
5	64.1 CH_2_	4.79 dd (5.1; 11.5)	C: 1, 3 Xyl1	
		3.87 dd (8.5; 11.5)		
Qui2 (1→2Xyl1)				
1	102.1 CH	5.18 d (8.4)	C: 2 Xyl1	H-2 Xyl1; H-3, 5 Qui2
2	**82.3** CH	3.93 t (9.0)	C: 1 Xyl5, C: 1, 3 Qui2	H-1 Xyl5
3	75.3 CH	3.99 t (9.0)	C: 2, 4 Qui2	
4	**86.4** CH	3.46 t (9.0)	C: 1 Glc3; C: 3, 5 Qui2	H-1 Glc3; H-2 Qui2
5	70.8 CH	3.57 dd (6.4; 9.0)	C: 4 Qui2	H-1 Qui2
6	17.8 CH_3_	1.57 d (6.4)		
Glc3 (1→4Qui2)				
1	104.0 CH	4.78 d (7.5)	C: 4 Qui2	H-4 Qui2; H-3, 5 Glc3
2	73.5 CH	3.83 t (8.9)	C: 1, 3 Glc3	
3	**85.9** CH	4.18 t (8.9)	C: 1 MeGlc4; C: 2, 4 Glc3	H-1 MeGlc4; H-1, 5 Glc3
4	69.0 CH	3.87 t (8.9)	C: 3, 5, 6 Glc3	
5	75.2 CH	4.05 m		H-1 Glc3
6	*67.1* CH_2_	4.94 d (10.3)		
		4.62 dd (6.2; 10.3)	C: 5 Glc3	
MeGlc4 (1→3Glc3)				
1	104.4 CH	5.21 d (8.2)	C: 3 Glc3	H-3 Glc3; H-5 MeGlc4
2	74.5 CH	3.85 t (8.2)	C: 1, 3 MeGlc4	
3	87.0 CH	3.67 t (8.2)	OMe; C: 2, 4 MeGlc4	H-1 MeGlc4; OMe
4	70.3 CH	3.89 m	C: 5 MeGlc4	
5	77.5 CH	3.89 m	C: 4 MeGlc4	H-1 MeGlc4
6	61.7 CH_2_	4.34 d (11.7)		
		4.05 brd (11.7)	C: 4, 5 MeGlc4	
OMe	60.6 CH_3_	3.80 s	C: 3 MeGlc4	
Xyl5 (1→2Qui2)				
1	105.1 CH	5.22 d (7.5)	C: 2 Qui2	H-2 Qui2; H-3, 5 Xyl5
2	74.9 CH	3.92 t (7.5)	C: 1 Xyl5	
3	76.4 CH	4.06 t (7.5)	C: 2, 4 Xyl5	
4	70.1 CH	4.04 m		
5	66.4 CH_2_	4.28 dd (4.8; 11.7)	C: 1, 3, 4 Xyl5	
		3.65 brdd (9.0; 11.7)	C: 3, 4 Xyl5	H-1 Xyl5

^a^ Recorded at 125.67 MHz in C_5_D_5_N/D_2_O. ^b^ Recorded at 500.12 MHz in C_5_D_5_N/D_2_O. ^c^ Bold—interglycosidic positions. ^d^ Italics—sulfate positions. Multiplicity by 1D TOCSY. The original spectra of **8** are provided in Appendix A.

**Table 10 marinedrugs-21-00602-t010:** The cytotoxic activities of glycosides **1**–**10**, djakonovioside A_1_, and cisplatin (positive controls) against human erythrocytes and MCF-10A, MCF-7, T-47D, and MDA-MB-231 human cell lines.

Glycosides	ED_50_, µM, Erythrocytes	Cytotoxicity, IC_50_ µM			
		MCF-10A	MCF-7	T-47D	MDA-MB-231
djakonovioside C_1_ (**1**)	8.55 ± 0.04	21.79 ± 0.72	38.64 ± 2.36	16.82 ± 0.76	7.67 ± 0.32
djakonovioside D_1_ (**2**)	7.74 ± 0.37	>50.0	>50.0	>50.0	>50.0
djakonovioside E_1_ (**3**)	21.67 ± 0.94	>50.0	1.52 ± 0.14	>50.0	2.19 ± 0.17
djakonovioside F_1_ (**4**)	0.51 ± 0.01	23.21 ± 0.78	36.75 ± 0.73	22.53 ± 1.78	20.04 ± 0.63
okhotoside A_2_-1 (**5**)	1.53 ± 0.14	12.16 ± 0.16	39.81 ± 0.18	11.86 ± 0.42	12.6 ± 0.16
cucumarioside A_2_-5 (**6**)	1.63 ± 0.13	14.02 ± 1.03	25.68 ± 1.49	5.81 ± 0.86	2.58 ± 0.10
frondoside A_2_-3 (**7**)	>50.0	>50.0	>50.0	>50.0	>50.0
cucumarioside A_3_-2 (**8**)	17.91 ± 0.48	>50.0	>50.0	>50.0	>50.0
isokoreoside A (**9**)	17.44 ± 0.57	>50.0	>50.0	>50.0	>50.0
koreoside A (**10**)	>50.0	>50.0	>50.0	>50.0	>50.0
djakonovioside A_1_	2.52 ± 0.23	17.51 ± 0,54	26.43 ± 0.14	15.25 ± 0.96	12.05 ± 0.54
cisplatin	-	77.31 ± 1.32	109.42 ± 2.01	>160.0	76.60 ± 1.20

**Table 11 marinedrugs-21-00602-t011:** Tumor cell selectivity index (SI; a ratio of IC_50_ calculated for healthy and cancer cells) of tested glycosides.

No.	Glycoside	Selectivity Index (SI)		
		MCF-7	T-47D	MDA-MB-231
**1**	djakonovioside C_1_	0.56	1.34	2.84
**3**	djakonovioside E_1_	>32.89	-	>22.83
**5**	okhotoside A_2_-1	0.31	1.03	0.97
**4**	djakonovioside F_1_	0.63	1.02	1.16
**6**	cucumarioside A_2_-5	0.55	2.41	5.43
	djakonovioside A_1_	0.66	1.15	1.45
	cisplatin	0.71	<0.48	1.01

## Data Availability

The original data presented in the study are included in the article/Appendix A.

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
