# Peer review of "Sulfated Triterpene Glycosides from the Far Eastern Sea Cucumber Cucumaria djakonovi: Djakonoviosides C1, D1, E1, and F1; Cytotoxicity against Human Breast Cancer Cell Lines; Quantitative Structure–Activity Relationships"

_marinedrugs, 2023, doi:10.3390/md21120602_

Round 1

Reviewer 1 Report

Comments and Suggestions for Authors

In this manuscript, authors carried out a chemical investigation on the Far Eastern sea cucumber Cucumaria djakonovi, resulting in the discovery of four new mono- and trisulfated triterpene penta- and tetraosides, together with six known derivatives. Their structures were established by the extensive analysis of NMR data and comparisons with the related known compounds. Possible biogenetic relationships of these glycosides were proposed. Moreover, cytotoxic bioassay results disclosed the different levels of cytotoxic activities for these compounds, and some of which was studied in further bioassays. Finally, QSAR revealed the extremely complex nature of such relationships, but these calculations correlated well with the observed SAR. Undoubtedly, these findings were important, which made this work worth publishing in this journal.

However, minor revisions were required.

1. P3L114: ‘S17–22’ → ‘S17–S22’

2. Table 1: ‘dd’ or ‘d’ for the data ‘4.28 d (5.6; 12.0)’? Please check it.

3. Please check the mass data for the fragment ion-peaks of new compounds, because not all data were exactly consistent with the observed in the mass spectra, such as ‘1223.4’(P5L150) and ‘1355.5’(P8L190).

4. P6L166: ‘with 1H,1H COSY spectrum’ → ‘in the 1H,1H COSY spectrum’

5. Why are the protons of methylene group CH2-23 adjacent to 22-oxo group and 24(25)-double bond easily exchanged to deuterium when the compound 4 is dissolved in the mixture C5D5N/D2O? How about the protons for the steroids possessing the same side chain?

6. P9L256: Were both protons of methylene group CH2-12 displayed the NOE correlations with the protons of methyl group CH3-21?

7. P10L274: Please add the number ‘9’ for  the compound ‘isokoreoside A’.

8. P12L293: ‘known earlier compounds’ → ‘known compounds’

9. P12L304: ‘Cucumaria conicospermium’ → ‘C. conicospermium

10. Please add the missing unit for the ED50/IC50 data in the maintext on P13 and P14.

Comments on the Quality of English Language

There were a few typo errors. Some of them were given in the comments to the authors.

Author Response

We are very grateful to the Reviewer for so carefull and professional cheking of the manuscript. We are fully accept the comments, and we hope to give complete answers to the questions.

  1. P3L114: ‘S17–22’ → ‘S17–S22’ fixed
  2. Table 2: ‘dd’ or ‘d’ for the data ‘4.28 d (5.6; 12.0)’? Please check it. Fixed, of course the multiplicity of hydroxy methylene group is “dd”.
  3. Please check the mass data for the fragment ion-peaks of new compounds, because not all data were exactly consistent with the observed in the mass spectra, such as ‘1223.4’(P5L150) and ‘1355.5’(P8L190). The value “1223.4” corrected to “1223.5” in the text of manuscript, corresponding to the observed ion-peak, as well as the values “1355.5” and “677.2” corrected to “1355.6” and “677.3”, correspondingly.
  4. P6L166: ‘with 1H,1H COSY spectrum’ → ‘in the1H,1H COSY spectrum’             fixed
  5. Why are the protons of methylene group CH2-23 adjacent to 22-oxo group and 24(25)-double bond easily exchanged to deuterium when the compound 4is dissolved in the mixture C5D5N/D2O? How about the protons for the steroids possessing the same side chain?             The easy exchange of the protons of C-23 group occurs due to the significant inductive effects of the neighboring oxo-group and double bond those shift the electron density of these protons making them available for exchange. Moreover, the dipole molecules of water greatly facilitate the exchanging of protons. The replacement of D2O to H2O in the mixture with C5D5N that used as the solvent for NMR spectra acquiring resulted in the appearance of the signal of C-23 in the 13C NMR spectrum, that is clearly evidences the exchange occurs between water and H2-23. As concerns the steroids, we do not find the compounds having exactly the same side chain in the structural databases. Moreover, the NMR spectra of sterols are usually acquired in CDCl3. So, we cannot compare the NMR spectra.
  6. P9L256: Were both protons of methylene group CH2-12 displayed the NOE correlations with the protons of methyl group CH3-21? The signal of H-12β at dH 1.97 demonstrated NOE-correlation with H-21 (the corresponding cross-peak is added to Table 4), while the site of cross-peak of H-12α at dH 1.76 and H-21 falls under the base line of the ROESY spectrum. The analysis interatomic distances in MM2 optimized model of the aglycone of 4 showed the last correlation would not be observed due to the distance between H12α and the closest proton of methyl group CH3-21 being more than 3.10 Å, while the distance between H-12β and H-21 is 2.19 Å.
  7. P10L274: Please add the number ‘9’ for  the compound ‘isokoreoside A’. fixed
  8. P12L293: ‘known earlier compounds’ → ‘known compounds’ fixed
  9. P12L304: ‘Cucumaria conicospermium’ → ‘C. conicospermiumdone

10. Please add the missing unit for the ED50/IC50 data in the maintext on P13 and P14.    units for the ED50/IC50 are added

Reviewer 2 Report

Comments and Suggestions for Authors

I find this manuscript as the excellent quality work, sound and modern techniques, interesting new compounds, interesting structure-activity relationship studies.

Detailed comments: 

Please decribe on how sugar configutaion was established

Line 649 and over the text: do not begin sentence with numbers 

Comments on the Quality of English Language

English i clear and acceptable

Author Response

We are very appreciative to the Reviewer for such high assessment of our work. Here are the replies to the comments:

Please describe on how sugar configuration was established.                               The sentence:The sugar configurations in the glycosides 14 were assigned as D on the basis of analogy with all other known triterpene glycosides from sea cucumbers.” was added to the manuscript.

In all the cases when absolute configuration of monosaccharide residues was determined by GLC of corresponding peracetates of aldononitriles or by GLC of acetates of octyl-derivatives of sugars it was established as D, even when the unusual sugars (as for example 3-O-methylglucuronic acid in the glycoside from the sea cucumber Synapta maculata) were found. The biogenetic background and the presence of common glycosides in different species of the sea cucumbers allowed to extrapolate the conclusion about D-configuration to all the representatives of the class Holothuroidea.

Line 649 and over the text: do not begin sentence with numbers.                 fixed

Reviewer 3 Report

Comments and Suggestions for Authors

The manuscript „Sulfated Triterpene Glycosides from the Far Eastern Sea Cucumber Cucumaria djakonovi: Djakonoviosides C1, D1, E1 and F1; Cytotoxicity Against Human Breast Cancer Cell Lines; Quantitative Structure – Activity Relationships” was submitted to Marine Drugs for publication.

Broad comments:

In the present study, ten triterpene glycosides were isolated from the sea cucumber Cucumaria djakonovi, of which four were identified as previously undescribed natural products. Additionally, cytotoxic potential of the isolates was assessed, Djakonovioside E1 and Cucumarioside A2-5 representing the most promising candidates.

The authors show a profound knowledge of structure elucidation, especially with regard to the complex glycosylation patterns to be resolved. Also, the isolation of the compounds is sufficiently documented as is the activity evaluation.

The only major comment from my side is the – in my opinion – inadequate introduction. With respect to the readers less familiar with marine organism it would be helpful to provide some information on sea cucumbers and their metabolome as well as their ecological role in the marine environment.

Furthermore, the style of the introduction does not necessarily appear as such but more like the end of a discussion or a summary, respectively.

Another point with regard to the introduction but also to the manuscript in general is that quite a high ratio of cited literature was written by the authors themselves. Is there only restricted literature available or is the topic of such exclusivity?

Minor comments:

Please provide IR spectra and melting points of the isolated compounds.

Author Response

We are gratefull for thorough checking of our manuscript and we agree with all the comments, so the text was improved in accordance to Reviewer's comments:

The only major comment from my side is the – in my opinion – inadequate introduction. With respect to the readers less familiar with marine organism it would be helpful to provide some information on sea cucumbers and their metabolome as well as their ecological role in the marine environment. Furthermore, the style of the introduction does not necessarily appear as such but more like the end of a discussion or a summary, respectively.

The introduction was changed in accordance to the Reviewer 3 comments: it was broadened with the information concerning the sea cucumbers habitat and ecological role, as well as with the discussion of studies of metabolic glycosides profiles. The corresponding references were added and the ordinal numbers of the next references were changed. The style of introduction was also corrected to became more suitable for this part of the paper.

Another point with regard to the introduction but also to the manuscript in general is that quite a high ratio of cited literature was written by the authors themselves. Is there only restricted literature available or is the topic of such exclusivity?

High level of self-citation is actually explained by the fact that our research team is a leading group in the world dealing with isolation and structure elucidation of triterpene glycosides from the sea cucumbers. Nevertheless, only necessary self-citations describing the compounds with the same structural elements required for identification or comparisons were provided. All the appropriate references to the other authors investigating different aspects of sea cucumber secondary metabolites were cited. The additional references to the metabolomic researches conducted by the other scientists also were added to the manuscript.

Minor comments:

Please provide IR spectra and melting points of the isolated compounds.

From the IR spectra the information concerning the presence of hydroxyls, carbonyl and sulfur can be only obtained. So, this method would show no differences between the structures of investigated glycosides because all of them share these structural features. That is why this method is not informative and more suitable precise and modern methods of the structure elucidation, such as NMR spectroscopy allowing to evidence the presence and positions of these functional groups based on the chemical shift values and high-resolution mass spectrometry that gave exact mass of the compounds were applied. Moreover, due to the small quantities of isolates were obtained we would prefer to save them for further experiments on biological activities but not IR spectra registration.

Melting points of the glycosides were provided.